# Mechanical Brain Injury Increases Cells’ Production of Cystathionine β-Synthase and Glutamine Synthetase, but Reduces Pax2 Expression in the Telencephalon of Juvenile Chum Salmon, *Oncorhynchus keta*

**DOI:** 10.3390/ijms22031279

**Published:** 2021-01-28

**Authors:** Evgeniya V. Pushchina, Eva I. Zharikova, Anatoly A. Varaksin

**Affiliations:** A.V. Zhirmunsky National Scientific Center of Marine Biology, Far East Branch, Russian Academy of Sciences, 690041 Vladivostok, Russia; eva_1213@mail.ru (E.I.Z.); anvaraksin@mail.ru (A.A.V.)

**Keywords:** adult neurogenesis, traumatic brain injury, glutamine synthetase, aNSCs, NPCs, radial glia, cystathionine β-synthase, Pacific chum salmon, Paired Box2, sonic hedgehog signaling

## Abstract

The considerable post-traumatic brain recovery in fishes makes them a useful model for studying the mechanisms that provide reparative neurogenesis, which is poorly represented in mammals. After a mechanical injury to the telencephalon in adult fish, lost neurons are actively replaced due to the proliferative activity of neuroepithelial cells and radial glia in the neurogenic periventricular zone. However, it is not enough clear which signaling mechanisms are involved in the activation of adult neural stem cells (aNSC) after the injury (reactive proliferation) and in the production of new neurons (regenerative neurogenesis) from progenitor cells (NPC). In juvenile Pacific salmon, the predominant type of NSCs in the telencephalon are neuroepithelial cells corresponding to embryonic NSCs. Expression of glutamine synthetase (GS), a NSC molecular marker, was detected in the neuroepithelial cells of the pallium and subpallium of juvenile chum salmon, *Oncorhynchus keta*. At 3 days after a traumatic brain injury (TBI) in juvenile chum salmon, the GS expression was detected in the radial glia corresponding to aNSC in the pallium and subpallium. The maximum density of distribution of GS^+^ radial glia was found in the dorsal pallial region. Hydrogen sulfide (H_2_S) is a proneurogenic factor that reduces oxidative stress and excitotoxicity effects, along with the increased GS production in the brain cells of juvenile chum salmon. In the fish brain, H_2_S producing by cystathionine β-synthase in neurogenic zones may be involved in maintaining the microenvironment that provides optimal conditions for the functioning of neurogenic niches during constitutive neurogenesis. After injury, H_2_S can determine cell survivability, providing a neuroprotective effect in the area of injury and reducing the process of glutamate excitotoxicity, acting as a signaling molecule involved in changing the neurogenic environment, which leads to the reactivation of neurogenic niches and cell regeneration programs. The results of studies on the control of the expression of regulatory Sonic Hedgehog genes (Shh) and the transcription factors Paired Box2 (Pax2) regulated by them are still insufficient. A comparative analysis of Pax2 expression in the telencephalon of intact chum salmon showed the presence of constitutive patterns of *Pax2* expression in neurogenic areas and non-neurogenic parenchymal zones of the pallium and subpallium. After mechanical injury, the patterns of *Pax2* expression changed, and the amount of Pax2^+^ decreased (*p* < 0.05) in lateral (Dl), medial (Dm) zones of the pallium, and the lateral zone (Vl) of the subpallium compared to the control. We believe that the decrease in the expression of Pax2 may be caused by the inhibitory effect of the Pax6 transcription factor, whose expression in the juvenile salmon brain increases upon injury.

## 1. Introduction

Active brain recovery after injury observed in fishes provides a useful model for studying the mechanisms responsible for reparative neurogenesis, while mammals exhibit a reduced neurogenesis capacity. It has been reported that in fish brain the generation of functional neurons from precursors (neurogenesis) and activated stem cells, or transdifferentiation of key elements, occurs during regeneration [1,2]. In contrast, the efficiency of neurogenic processes in various vertebrates is significantly reduced in some cases [3]. The lost neurons were actively replaced through the proliferative activity of neuroepithelial cells and radial glia in the periventricular zone after a mechanical injury to the telencephalon in zebrafish *Danio rerio* [4], the catfish *Scyliorhinus canicula* [5], and masu salmon *Oncorhynchus masou* [6]. However, it still remains unclear what signaling mechanisms are involved in the activation of adult neural stem cells after damage (reactive proliferation) and in the production of new neurons (regenerative neurogenesis) from progenitor cells.

The central nervous system (CNS) in fishes and amphibians has the highest capacity for neurogenesis, with the physiological neurogenesis and transdifferentiation of pre-existing elements launched simultaneously after a traumatic injury [7]. The physiological and regenerative neurogenesis in reptiles has not yet been sufficiently elucidated [8]. However, the results of studies on the medial cortex in the leopard gecko *Eublepharis macularius* show the presence of proliferating pools of neural stem/progenitor cells in the *septomedialis sulcus*, a ventricular zone adjacent to the medial cortex. These cells express sex determining region Y-box 2 (SOX2), glial fibrillar acidic protein (GFAP), and vimentin, and have a radial glial morphology [9]. Nerve tissue can be partially or completely regenerated in certain areas of the brain in cartilaginous fish. For example, in studies on the catshark *Scyliorhinus canicula*, various proliferating populations of radial glia cells were found in the pial zone [5]. However, the cells of the proliferating radial glia (RG) that can act as neurogenic and/or gliogenic precursors have not been identified.

It has been shown that in fish, both in case of induced inflammation and in traumatic brain injury, the proliferation of progenitor cells and, as a consequence, the number of new neurons, increases [10]. This suggests that the inflammatory response is one of the molecular signals that inevitably precede the activation of adult neural stem cells (aNSCs) [3,7]. This process, observed after CNS injury, is triggered by the activation of injury-induced molecular programs [11].

Studies on mammals have shown that acute inflammation interferes with adult neurogenesis and regenerative processes [3]. Thus, it is obvious that in fishes, in contrast to mammals, the inflammation is a positive regulator of neuronal regeneration in CNS. A rapid development of the active inflammatory response after injury was recorded based on the increased expression of proinflammatory cytokines such as IL-8, IL-1b and tumor necrosis factor [10]. These data suggest that an injury to the fish brain leads to the acute inflammatory response, which is necessary for the enhancement of the neuron progenitor cell (NPC) proliferation and subsequent neurogenesis. Thus, it is likely that the acute inflammatory response in fish provides a special microenvironment on the cellular and molecular levels where molecular programs for regenerative neurogenesis are initiated. After creating the special microenvironment, the inflammatory process is eliminated within a few days without signs of chronic inflammation such as those observed in the mammalian brain [12].

Several key substances induce neurogenesis and reparative processes, with H_2_S being a candidate for the regulation of intercellular interactions performing the common cytoprotective and signal functions [13,14]. It is a neuromodulator considered to be similar in functions to other gaseous compounds such as nitrogen oxide (NO) and carbon monoxide (CO). H_2_S is produced by several pathways, in particular by cystathionine β-synthase (CBS) which is a key enzyme in the formation of cysteine from methionine. This gasotransmitter activates intracellular signaling pathways through the sulfhydration of proteins and can react with protein hemes. There are multiple interplays between H_2_S and other gasotransmitters both on the level of their generation and as targets [15]. Previous studies on traumatic injury to the cerebellum in juvenils of masu salmon *Oncorhynchus masou* demonstrated sharply increasing the number of CBS^+^ cells after 3 days, which indicates the involvement of H_2_S in the post-traumatic response [16]. Similar results were observed after optic nerve injury in trout, which showed a significant increase in the number of H_2_S-producing cells in the integrative centers of the brain: telencephalon, optic tectum, and cerebellum [17]. A noteworthy finding was the presence of CBS-producing radial glia in the optic tectum of trout after the optic nerve injury [18]. H_2_S has a significant effect on physiological and pathophysiological processes in the CNS, being involved in protective mechanisms induced by traumatic brain injury (TBI) and ischemic reperfusion [19]. Acute inflammation in mammals typically has a negative effect on neurogenesis and regeneration by promoting glial scar formation and inhibiting the proliferation of progenitor cells, as well as the migration, survival, maturation, and integration of new neurons [20,21]. The inflammatory response observed at the site of injury in the fish brain after TBI, on the contrary, did not limit neuronal regeneration [3,12].

Currently, the involvement of H_2_S in the processes of ischemic brain injury, TBI and the involvement of this gas transmitter in the control of oxidative stress and the increase in reactive oxygen species in the H_2_S-dependent signaling are being actively studied [19,22,23]. H_2_S reactions with many signaling mediators, transcription factors, and channel proteins are known to occur in neurons and glial cells both in vivo and in vitro [14,19]. However, information on the intercellular interaction and the involvement of H_2_S in regenerative processes, in particular, in adult neurogenesis and TBI, is still limited.

Adult neurogenesis and neuronal regeneration after injury are controlled by the activation of certain molecular pathways, including transcription factors, growth factors, neurotrophins, and cytokines, which are expressed in certain neurogenic niches and, ultimately, at the damaged CNS site. Shh signaling plays an important role in both CNS neurogenesis and regeneration [24]. Transcription factors of the Paired Box (PAX) family are one of the factors regulated by the Shh signaling pathway; however, the mechanisms regulating the Pax2 expression are almost unstudied [25]. Studies on trout have shown that damage to the optic nerve leads to an increase in the number of Pax2^+^ reactive astrocytes in it, being involved in the initial stages of the optic nerve axon regeneration [26]. In the case of optic nerve injury, a significant increase in the number of Pax6+ cells has been revealed in the parts of the trout brain that have directed retinal inputs (the visual nuclei of the diencephalon and the optic tectum) [27]. It has been found that some of the Pax6+ cells have a neuroepithelial phenotype and are part of reactive neurogenic niches located in the periventricular zone (PVZ) and parenchymal regions of the brain. Another population of Pax6+ cells has a radial glia phenotype and arises as a result of activation of constitutive neurogenic domains, as well as within newly formed reactive neurogenic niches [26].

Juvenile Pacific salmon is a convenient model for the study of postembryonic neurogenesis, since the intensive neurogenesis in its brain is provided by neuroepithelial progenitors [6,16,28]. The brain of Pacific salmon largely retains the features of a primitive (ancestral) organization, while maintaining the high proliferative activity of neuroepithelial (NE) cells, which is characteristic of embryonic development. The high neurogenic activity in the juvenile masu salmon *O. masou* is largely due to the presence of tissue-specific precursors of the neuroepithelial type [6]. Such cells divide symmetrically, constantly increasing the pool of neural precursors, which subsequently form neurons, glia, and ependymal cells that make up the CNS structure. Similar properties especially actively develop during the first year of life of the embryonic NSCs in juvenile Pacific salmon, providing the fetalization process associated with developmental delay and the retention of features of the embryonic brain organization [29]. The main goal of the present work is to study the ratio of NE and RG precursors in the brain of juvenile salmonids and the involvement of both types of precursors in the reparative response in TBI, as well as to study the role of hydrogen sulfide in the development of the acute phase of post-traumatic response and the involvement of Pax2 in the regulation of this process. To achieve this goal, we analyzed the comparative distribution of glutamine synthetase (GS) as a molecular marker of NSCs, CBS, and transcription factor (TF) Pax2 in the telencephalon of intact juvenile chum salmon *Oncorhynchus keta* and in fish at three days after a mechanical injury.

## 2. Results

In the telencephalon of the juvenile chum salmon *Oncorhynchus keta*, the dorsal (D) and ventral (V) regions, corresponding to the pallial and subpallial regions of zebrafish, were identified [20]. The dorsal (DD), lateral (DL), central (DC) and medial (DM) zones were distinguished within the pallial region. In the subpallial region, the dorsal (VD) and ventral (VV) nuclei and the lateral (VL) zones were found.

### 2.1. Glutamine Synthetase in the Telencephalon of Intact Chum Salmon

The patterns of the GS labeling in the telencephalon of the intact animals related to the matrix periventricular zones (Figure 1A). In the pallium of juvenile chum salmon, GS labeling in intact animals was revealed in some periventricular zone (PVZ) cells, forming areas of up to 300 μm in length as a discontinuous monolayer, or including small aggregations of immunopositive cells (Figure 1B). GS labeling was found also in small, intensely labeled neuroepithelial cells of rounded or oval shape and devoid of processes (Figure 1B, Appendix A). In the subventricular zone (SVZ), single GS-labeled cells were identified, and in the deeper parenchymal zone layers (PZ), rare intensely and moderately labeled GS cells were found (Figure 1B, Appendix A). Few intensely labeled cells of larger size were found in DC (Figure 1A,B).

In DD, small, intensely labeled GS^+^ cells of the neuroepithelial type were located above the layer of immunonegative cells in the PVZ (Figure 1C, Appendix A). In SVZ, there were single small cells with moderate GS labeling, occasionally forming small clusters (Figure 1C, inset, Appendix A). In the deeper layers (PZ), heterogeneous cell complexes were identified, including immunonegative cells with contacts with clusters of small, moderately labeled GS^+^ cells (Figure 1C). In DD, the number of GS^+^ cells was relatively high (Figure 1J).

DM was characterized by the presence of immunopositive granules and more intensely labeled NE cells organized into small groups, alternating with immunonegative areas less extensive than those in DD (Figure 1D, Appendix A); some of the cells penetrated into SVZ (Figure 1D, dotted inset). In PVZ, we identified small clusters of GS-labeled cells forming dense, surface, spherical formations in the neuroepithelium (Figure 1D, dotted inset), with dense groups of small, intensely labeled cells of irregular shape located along them (Figure 1D, inset in a black rectangle). In the deeper (PZ) layers of DM, we also identified few, intensely labeled rounded cells (Figure 1D, Appendix A). In DM, the number of GS^+^ cells was the lowest among all pallial zones (Figure 1J).

In DL, single small cells located in the basal part of PVZ were found (Figure 1E, Appendix A). The distribution patterns of small, moderately labeled GS cells and cytoplasm of NE cells surrounding immunonegative nuclei often prevailed in PVZ (Figure 1E, inset). In the deep layers of PZ, oval and small, moderately labeled GS cells and numerous local clusters of GS-negative cells were found, which created constitutive neurogenic niches (CNN) of parenchymal localization (Figure 1E). In DL, the number of GS^+^ cells was at its maximum (Figure 1J).

In the subpallial region of the juvenile chum salmon telencephalon, GS was detected within intensely labeled cell groups in VD (Figure 1F, Appendix A), as well as in the apical part of PVZ in single cells alternating with GS-negative neurons (Figure 1F, Appendix A). Small, moderately and intensely GS^+^ cells were distributed within the VD cell aggregations (Figure 1F, Appendix A).

VV contained single or paired moderately labeled cells in the apical part of PVZ (Figure 1G, Appendix A). Solitary, intensely labeled elongated cells had poorly developed processes (Figure 1G, Appendix A). Another type of cell of a pseudo-unipolar form with an immunonegative nucleus was found in SVZ: they penetrated into the deeper layers of PZ and, apparently, also belonged to the migrating population (Figure 1G, Appendix A). Paired, moderately labeled, oval GS^+^ cells were localized in the surface of PVZ (Figure 1G, Appendix A). In SVZ, aggregations of intensely labeled granule-like subcellular elements, ranging in size from 8.5 to 18 µm, were revealed (Figure 1G,F, Appendix A).

In VL, single small, intensely labeled cells in PVZ were visualized; in the deeper layers of PZ, intensely labeled clusters and rare larger cells were identified (data not shown, Appendix A). Moderately labeled granules and two types of cells were present in all the zones (Appendix A).

An analysis of the densitometric activity of GS showed that there were significant (*p* < 0.05, *p* < 0.01) differences between the groups of intensively, moderately labeled cells and immunonegative cells in the telencephalon of juvenile chum salmon (Figure 1H). The percentage ratio of GS-immunopositive and negative cells in all areas of the pallium and subpallium is shown in Figure 1I.

### 2.2. GS Expression in the Telencephalon of Chum Salmon on Day 3 Post-Injury

After traumatic injury, GS^+^ cells with the radial glia phenotype were recorded to appear in the telencephalon (Figure 2A, Appendix A). For a detailed study of the RG distribution in the pallium and subpallium of juvenile chum salmon, the zones of the dorsal and ventral regions of the telencephalon were subdivided into subregions (Figure 2B, Appendix A). In each of the pallium zones (DD, DM, and DL), an additional three subzones were identified to characterize the distribution of radial glial cells corresponding to aNSCs of other vertebrates. In intact individuals, the predominant types of GS^+^ cells were NE cells. We believe that the GS^+^ activation of the radial glia occurred as a result of the traumatic injury, which should be considered as a post-traumatic response in juvenile chum salmon of this age group and may be species-specific.

At three days post-injury, the patterns of RG distribution in the pallium were accompanied by the intensive cell migration and emergence of additional reactive neurogenic niches (RNN) (Figure 2C). RG after injury was detected at different distances from PVZ and was represented by both single fibers and bundles of GS^+^ fibers (Figure 2D). In the area of DD, located next to DM (designated as DD1), RNNs of heterogeneous cellular composition were observed to emerge after injury (Figure 2E). In some cases, discrete GS-labeled areas containing small reactive, intensely labeled GS^+^ clusters or single cells were recorded in PVZ (Figure 2E, Appendix A).

Populations of GS^+^ cells in such areas differed in morphological heterogeneity and intensity of GS labeling (Figure 2E, Appendix A). Cells without radial processes, moderately or intensely GS-labeled, were located in the basal part of PVZ (Figure 2E, Appendix A). Thin GS^+^ fibers had predominantly a diffuse distribution pattern (Figure 2E).

In DD2, intensely labeled RG cells had basal and apical processes, ending in PVZ with end-feet (Figure 2F). Such cells formed dense clusters in PVZ, and the bodies of GS^+^ cells often had contact with the processes of nearby cells (Figure 2F, Appendix A). Clusters of GS-neuroblasts were identified in SVZ and PZ (Figure 2F, Appendix A). In most cases, the radial processes of GS^+^ cells extended for a considerable distance into the deep layers of DD (Figure 2E,F). In DD2, very large aggregations of GS-neuroblasts were revealed in PZ, to which the radial fibers from SVZ approached (Figure 2F). Obviously, the patterns of neuroblast migration along RG outgrowths were directed to the area of injury. Our observations in DD3 showed a correlation between the size of RNNs and the distribution pattern of RG fibers: larger bundles of RG fibers emerged from larger RNNs, (Figure 2G). Separate moderately labeled GS^+^ cells were identified along some bundles of RG outgrowths (Figure 2G, Appendix A).

Along with the newly appeared GS^+^ radial glia in all DD subregions, we identified a heterogeneous population of densely stained cells without processes localized in the basal part of the PVZ (Figure 2E–G, Appendix A). We believe that such cells were NE, or they represented a population of cells produced through asymmetric mitosis, which retracted the outgrowths by somal translocation or lost them by resorption.

After injury, the intensity of GS immunolabeling significantly increased in both intensely labeled and moderately labeled cells (*p* < 0.05), compared with control animals (Figure 2H). More intense optical density was also detected in negative cells (Figure 2H). In the post-traumatic period, significant hypertrophy of GS–neuroepithelium in PVZ and SVZ was observed in DD. Such cells were intensely stained with MG and had a high index of nuclear–cytoplasmic ratios (Figure 2F–G, Appendix A). Cell cytoplasm was characterized by high basophilia. A comparison of distributions of RG in the DD1, DD2, and DD3 regions showed significant intergroup differences between DD1 and DD2, as well as between DD1 and DD3 (*p* < 0.05); the maximum number of fibers was found in DD3; the minimum, in DD1 (Figure 3I).

In the medial part of the pallium (DM), three subzones, DM1, DM2, and DM3, were also identified (Figure 3A–C). In DM1, the diffuse pattern of GS^+^ distribution of radial glia persisted (Figure 3A, Appendix A). PVZ contained densely or moderately labeled GS^+^ cells of the neuroepithelial type (Figure 3A, Appendix A). In PVZ, DM2, small GS^+^ RNNs of the neuroepithelial type, separated by immunonegative regions, were identified (Figure 3B). In some areas of DM2, the typical RNN structure was not clearly expressed, and, instead, it was represented by separate or paired GS^+^ cells creating an intermittent layer (Figure 3B). In DM1 and DM2, the structure of SVZ and PZ formed by aggregations of GS cells was stratified (Figure 3A,B, Appendix A). In some areas of DM2 in SVZ, GS^+^ neuropil and GS^+^ cells migrating along RG fibers could be observed (Figure 3B, inset). The migration patterns along the GS + RG were most clearly seen in DM3 (Figure 3C). A comparison of distributions of GS + RG in DM showed significant intergroup differences between DM1 and DM3 (*p* < 0.05); the maximum number of fibers was localized in DM3; the minimum, in DM1 (Figure 3G).

In DL, GS^+^ cells generated a monolayer structurally similar to that in DD, and, thus, DL was also divided into three subregions: DL1, DL2, and DL3 (Figure 3D–F). In some cases, GS^+^ radial glia were clearly visible in DL, but the typical RNN structure was not found (Figure 3D, Appendix A). In DL1, the largest and longest RG fibers, organized into bundles, were identified (Figure 3D). In the post-traumatic period in DL1, the density of distribution of GS cells in SVZ and deeper PZ increased significantly (Figure 3D,H). Along with radially oriented patterns, DL2 contained patterns of tangentially oriented GS^+^ fibers, with extensive migration of a heterogeneous cell population observed along them (Figure 3F, inset). DL2 was characterized by the presence of GS^+^ neuroepithelial cells (Figure 3F, Appendix A). In DL3, a dense population of GS + RG with long outgrowths, penetrating deeply into PZ, was identified (Figure 3F, inset). A comparison of distributions of GS + RG in DL showed significant intergroup differences between DL1 and DL3 (*p* < 0.05); the maximum number of RG fibers was found in DL1; the minimum, in DL3 (Figure 3H).

In the subpallial region, the distribution patterns of GS + RNN in VD1 containing immunopositive RG somewhat resembled that in DD (Figure 4A, inset). In particular, numerous neuroepithelial cell clusters were identified in PVZ and SVZ (Figure 4A, inset), and the amount of RG was reduced compared to DD. Migrating immunopositive cells were found in the ventral part of the dorsal nucleus (VD2) along GS + RG fibers (Figure 4B, inset, Appendix A). In some cases, areas containing elongated GS^+^ cells were found in PVZ of VD2, with a layer of GS–neuroepithelial cells observed above (Figure 4B, Appendix A).

In VL, two subregions were identified: dorsal (VL1) and ventral (VL2) (Figure 4C,D). In VL1, aggregations of GS^+^ neuroepithelial cells dominated PVZ (Figure 4C, inset), and the amount of RG was very small. In VL2, single GS^+^ neuroepithelial cells were detected in PVZ; RG fibers were not found (Figure 4D, inset).

In the ventral area of the subpallium, two zones were also distinguished: dorsal (VV1) and ventral (VV2) (Figure 4E,F). RNNs of a mixed type were found in VV1, containing large densely stained GS^+^ groups of neuroepithelial cells, RG fibers, and single moderately labeled cells of parenchymal localization (Figure 4F, Appendix A). In VV1, large aggregations of immunonegative cells were found in SVZ (Figure 4F, inset, Appendix A). In VV2, few immunolabeled RG fibers, single GS^+^ aggregations of neuroepithelial cells in PVZ, and a high concentration of immunonegative cells in PZ were revealed (Figure 4F).

An analysis of the quantitative distribution of GS^+^ cells in the fish on day 3 post-injury showed significant differences in DD, VV, VL (*p* < 0.05), and DM (*p* < 0.01) from the control (intact) animals (Figure 4G). The percentage ratio between GS-immunopositive and negative cells in all areas of the pallium and subpallium after injury is shown in Figure 4H.

### 2.3. CBS Expression in the Telencephalon of Intact Chum Salmon

In the pallium, CBS labeling was detected in the DD, DL, DM, and DC zones (Figure 5A). The intensity of immunolabeling in cells was different, with two levels of immunostaining identified: high (4142 ± 35 UOD), at which nuclei and cytoplasm of cells were intensely stained, and moderate (2993 ± 28 UOD), where less intense labeling of the cell cytoplasm was observed (Figure 1B–D). Some of the cells, whose nuclei were stained with MG, were CBS-immunonegative (1430 ± 38 UOD). In the dorsal zone (DD) in PVZ, CBS labeling was detected in oval and/or small rounded cells (Figure 5B, Appendix A). In DM, some immunopositive cells showed proximal areas of radially directed processes (Figure 5C). Differences in the size and morphology of CBS^+^ cells were detected in DD (Appendix A); in DL and DM, immunopositive cells were morphologically similar (Appendix A). In DC, only moderately labeled rounded or oval cells were detected (Figure 5A).

The specific features of cell labeling in all zones of the dorsal region made it possible to distinguish both single immunolabeled cells and small CBS^+^ clusters in PVZ, alternating with extended areas that lacked CBS immunopositivity (Figure 5B). This pattern of distribution of immunopositive/immunonegative cells was typical for DD and DM (Figure 5B,C). In DL, a diffuse pattern of CBS immunolabeling was revealed; separate clusters of CBS^+^ cells were encountered in PVZ. The number of immunopositive cells in SVZ was limited, and the distribution of moderate CBS labeling in PVZ was homogeneous (Figure 5D). In general, in PVZ of the control animals, the CBS^+^ cells made up a monolayer in the dorsal zone (Figure 5B). The maximum density of distribution of intensely and moderately labeled cells was found in DD; in DL and DM, the values were somewhat lower (Figure 5F). The maximum number of CBS + cells was found in DD (Figure 5F).

In the subpallial region of intact brain, CBS immunopositivity was detected in the dorsal (VD), ventral (VV), and lateral zones (VL) (Figure 6A). The level of CBS activity in the cells was moderate and intense (Appendix A). In PVZ of VD, the number of immunopositive cells was limited, but the immunonegative cells often contained CBS^+^ granules (Figure 6B, inset, Appendix A).

CBS cells dominated SVZ, among which small aggregations of small elongated basophilic cells were revealed (Figure 6B, red inset, Appendix A). However, in the deeper PZ, there were aggregations of moderately and intensely homogeneously labeled CBS cells, single or organized into small clusters, as well as CBS–small basophilic cells (Figure 6B, Appendix A). In VV, CBS-immunopositive granules on the apical surface of PVZ neuroepithelial cells composed a thin, moderately labeled layer (Figure 6C, Appendix A). CBS^+^ granules were also found in single cells of PVZ, extending deep into SVZ (Figure 6C, inset). CBS^+^ small cells in PVZ formed few local clusters; in SVZ there were paired, moderately labeled small, elongated cells, but the patterns of CBS^+^ granular localization in cells were more characteristic (Figure 6C, Appendix A). In dense cell clusters of SVZ, single elongated moderately labeled CBS cells and granules were often found inside the cells and on the surface of the cells stained with MG (Figure 6C, inset). A few patterns of mitoses were encountered in the PZ area (Figure 6C, red inset). In VL, CBS^+^ granules were also detected on the surface of immunonegative NE cells (Figure 6D, Appendix A). Another type of CBS^+^ structure in PZ were clusters of small, intensely labeled cells, differing in distribution density and number (Figure 6D). In some cases, clusters of cells of a diffuse type were observed in PVZ and PZ (Figure 6D). Other clusters were denser and occurred in PVZ and SVZ (Figure 6D). In the deeper layers of PZ, multiple CBS^+^ granules were located on the surface of CBS–cells (Figure 6D, rectangle). One-way ANOVA data indicated differences in optical density between the groups of intensely and moderately labeled cells (*n =* 10), as well as immunonegative cells (Figure 6E). A comparative assessment of the number of intensely, moderately immunolabeled, and negative cells showed significant differences (*p* < 0.05, *p* < 0.01) between these groups (Figure 6F).

### 2.4. Distribution of CBS in the Telencephalon of Chum Salmon on Day 3 Post-Injury

After the injury to the telencephalon, the number of CBS-labeled cells in PVZ of the pallium and subpallium significantly increased compared to the control animals (Figure 7A and Figure 8A). The quantitative assessment of CBS^+^ cells in all areas of the pallium and subpallium at 3 days after the telencephalon injury showed a significant increase in the number of immunopositive cells in all areas of the pallium (DD, DM, DL) and the ventral nucleus of the subpallium (VV) (*p* < 0.05), as well as in VL (*p* < 0.01) (Figure 8A). In the ratio of the number of CBS^+^ intensely/moderately labeled cells in the intact animals, moderately labeled cells dominated (Figure 8B). However, in the acute post-traumatic period, this ratio changed significantly due to an increase in the number of intensely labeled cells (Figure 8C). A comparative study of the optical density level of intensely/moderately labeled cells in the control animals and in those exposed to injury showed significant intergroup differences (*p* < 0.05) in pallial and subpallial populations of intensely/moderately labeled cells (Figure 8D).

The pattern of distribution of CBS^+^ cells in DD, DM, and DL was represented by a multi-row layer of intensely labeled cells reaching a considerable length, interrupted at the border of the pallial zones (Figure 7B). CBS^+^ cells in the pallial areas were found separately or in a heterogeneously labeled layer containing cells of various sizes and morphologies (Figure 3C, Appendix A).

In the DM, small accumulations of CBS^+^ cells were recorded as a part of PVZ and SVZ (Figure 7C, inset); the number of labeled cells of parenchymal localization, located discretely or forming small clusters, increased (Figure 7C). In DD, the number of CBS^+^ cells in SVZ increased (Figure 3D), and the morphological heterogeneity of cells in this area also increased (Figure 7D, inset, Appendix A). In DL, the number of clusters containing intensely labeled cells in PVZ increased; clusters of CBS cells, including moderately CBS^+^ cells, appeared in SVZ (Figure 7F, Appendix A). Small clusters containing intensely labeled cells were recorded from PZ (Figure 7F).

In the subpallial region, similar rearrangements occurred in the dorsal (VD) and ventral (VV) nuclei (Figure 7F, inset, Appendix A). In PZ of the subpallium, the number of single and paired CBS^+^ cells, as well as their small clusters, increased (Figure 7F). The values of morphometric parameters of cells in the periventricular zone labeled with CBS differed slightly from those of the control animals (Appendix A). In general, in all areas of the pallial and subpallial areas after injury, cells of smaller sizes dominated (Appendix A).

### 2.5. Expression of the Transcription Factor Pax2 in the Telencephalon of Intact Chum Salmon

The expression patterns of the transcription factor (TF) Pax2 in the telencephalon of the intact animals were not directly related to the matrix periventricular zones and formed morphogenetic fields in which TF expression was present in nuclei and cells (Figure 9A, Appendix A). In the pallium, the patterns of Pax2 expression corresponded to the localization of the pallial regions: dorsal (DD), medial (DM), lateral (DL), and central (DC) (Figure 9A). At the border of the pallial and subpallial zones, an area of increased Pax2 expression separated the dorsal region from the ventral one (Figure 9A). The measurement of the optical density of Pax2 immunolabeling in the pallium and subpallium made it possible to distinguish intensively, moderately, and weakly labeled cells (Figure 9J, Appendix A).

In DD, the expression of Pax2 in PVZ was limited. The intensely labeled cells were superficially localized and formed small clusters separated by small immunonegative regions (Figure 9B, inset). Expression of Pax2 in DD cells was detected both in nucleus and in cytoplasm of cells (Appendix A). Both forms of Pax2 localization were identified in PVZ as part of CNN (Figure 9C, inset). In SVZ, individual nuclei of Pax2^+^ cells were moderately labeled (Figure 9B, Appendix A). In PZ, Pax2 expression was detected in nuclei that form small morphogenetic fields (Figure 9B, Appendix A). Along with Pax2^+^ nuclei, Pax2–cell clusters, as well as multidirectional immunonegative basophilic cell populations, were encountered in PZ (Figure 9B, Appendix A). In DM, the pattern of Pax2 expression in PVZ was present mainly in nuclei (Figure 9C, black inset, Appendix A), whose number in the PVZ was very high. Intensively labeled Pax2^+^ nuclei formed extensive areas in the territory of PVZ and SVZ (Figure 9C, inset). In PZ, the Pax2 expression patterns were diffused in the form of granules (Figure 9C, Appendix A). In DM, the radial direction of cell migration was clearly visible, with moderately/weakly labeled zones of Pax2 expression localized along it (Figure 9C).

In DL, on the territory of PVZ and SVZ, the number and morphological heterogeneity of Pax2^+^ cells was much higher than in DM and DD (Figure 9A, Appendix A). Most cells in the surface part of PVZ had nuclei labeled with Pax2; in the basal part of PVZ and SVZ, there were Pax2^+^ cells that formed extensive zones of TF expression (Figure 9D, Appendix A). In the parenchymal part of DL, both cellular and nuclear expressions of Pax2 were found (Figure 9D, Appendix A).

In VD, the immunolabeled cells in PVZ had a surface, tangential orientation (Figure 9F, inset, Appendix A). Intensely labeled Pax2 cells were few in number and usually formed small groups (Figure 9F, inset, Appendix A). In SVZ and in the parenchymal part of VD, there were single cells and nuclei moderately and weakly labeled with Pax2, forming morphogenetic fields of various lengths (Figure 9F, Appendix A). On the surface line, there were sparsely located Pax2^+^ nuclei in PVZ (Figure 9F, black inset, Appendix A). In SVZ, single intensely labeled cells, as well as local labeled parenchymal aggregations of Pax2^+^ cells, were identified (Figure 9F, red inset). In PZ, local Pax2-cell clusters of high distribution density were found, with dense clusters of intensely labeled Pax2^+^ cells adjoining them (Figure 9F, dashed rectangle, Appendix A). In VL, intensely labeled clusters of Pax2^+^ cells were detected in PVZ of various sizes and lengths (Figure 9G, inset, Appendix A). As a rule, several morphological types of Pax2^+^ cells were identified within the clusters (Figure 9G, black inset, Appendix A). Some of the cells migrated from PVZ into SVZ in large clusters. In the parenchymal part of VL, multidirectional immunonegative flows of cells and their accumulations, as well as a few moderately labeled Pax2 cells and nuclei, were observed (Figure 9G, Appendix A). In the area of PZ, there was an aggregation of immunonegative basophilic cells including single Pax2^+^ cells (Figure 9G, red inset).

The ratio of the number of Pax2^+^/Pax2– cells in the pallium and subpallium zones of the intact brain is shown in Figure 9H. One-way ANOVA was used to assess intergroup differences between DD/DM and DM/DL in the zones of pallium and VD/VV and VV/VL in the subpallium (*p* < 0.05). Significant differences between pallial (DD, DM, DL) and subpallial (VD, VV, VL) cell groups (*p* < 0.05) were revealed. A densitometric study of the optical density of Pax2^+^ immunolabeling in the pallium and subpallium showed significant differences between the intensely, moderately, poorly labeled cells and the immunonegative cells (Figure 9J).

### 2.6. Expression of Pax2 in the Telencephalon of Chum Salmon on Day 3 Post-Injury

On day 3 post-injury, intensely labeled Pax2 cells and nuclei localized in the PVZ and SVZ were identified in the dorsal part of the pallium (Figure 10A, inset). In the area of injury (Figure 10Aa), the cells located in the surface part of PVZ we considered as a population of cells migrating to the area of injury. Another population of Pax2-labeled cells formed local dense clusters in the basal part of PVZ and SVZ (Figure 10A,Aa, Appendix A). We found nuclei of Pax2-labeled cells located in deep layers of PVZ (Figure 10Aa). In the basal part of PVZ, separate or paired labeled cells were also found (Figure 10A, inset, Figure 10Aa). In the area of injury, aggregations of Pax2^+^ cells, tangentially migrating, intensely labeled Pax2 cells and nuclei were observed along the wound canal in DD (Figure 10Aa, in a red rectangle, Appendix A). In the parenchymal areas of the telencephalon adjacent to the injury zone, there was an increased density of distribution of Pax2^+^ intensely labeled cells compared with that in the control animals (Figure 10Aa,G, Appendix A). In DM, it was evident that a similar population of Pax2^+^ cells located in the surface areas of PVZ was a reactive population of cells migrating to the injury area (Figure 10B, inset, Appendix A). Another population of Pax2^+^ cells formed local dense clusters in the basal part of PVZ and SVZ (Figure 10B). We also found nuclei of Pax2^+^ cells located in deep layers of PVZ (Figure 10B, Appendix A). In DL, the expression pattern of Pax2^+^ cells largely corresponded to that in DD (Figure 10C, Appendix A). In particular, tangentially migrating cells were found in the apical and basal parts of PVZ (Figure 10C, inset). Separate Pax2^+^ cells were located around dense Pax2–clusters in PZ; sometimes there were single densely/moderately labeled cells in SVZ (Figure 10C, Appendix A). The density of distribution of Pax2–cells in the parenchymal areas in DL after injury significantly increased in comparison with the control (Figure 9K).

In the post-traumatic period, Pax2^+^ nuclei and superficial neuroepithelial cells were labeled in VD, some of which penetrated into the deeper layers of PVZ and SVZ (Figure 10D, black inset, Appendix A). The largest aggregations of Pax2^+^ cells were found in the superficial zone of the ventral part of the area and in SVZ of the central part of VD (Figure 10D, red inset). Another aggregation of Pax2^+^ cells was evenly distributed in PZ between these two clusters. In VL, the expression pattern of Pax2^+^ cells was similar to that of other subpallial regions; in the apical and basal parts of PVZ, the numbers of immunopositive cells and nuclei were increased (Figure 10F, inset). In SVZ and PZ, single moderately and weakly labeled Pax2 cells and nuclei prevailed (Figure 10E, Appendix A). In the ventral part of the nucleus, the distribution density of Pax2^+^ cells after injury was slightly increased (Figure 10G).

In VL, the distribution of Pax2^+^ cells and their aggregations in the surface and basal parts of PVZ was increased compared to the control (Figure 10F, inset). A similar distribution of cells was typical for other ventral areas; however, local areas containing numerous Pax2^+^ nuclei were identified in VL, in deeper layers of PVZ (Figure 10F, Appendix A). Small clusters of Pax2^+^ cells were relatively common in SVZ, while single moderately labeled Pax2 cells dominated PZ (Figure 10F, Appendix A).

The data of quantitative analysis showed that the number of Pax2^+^ cells significantly decreased (*p* < 0.05) in the medial and lateral areas of the pallium (DM, DL) and the lateral area of the subpallium (VL) after injury (Figure 10G) compared to the control. In the area of injury (DD), there was also a slight decrease in the number of Pax2^+^ cells (Figure 10G), but no significant differences from the control groups were found. The intensity of Pax2 immunolabeling also showed a tendency to decrease, but there were no significant differences in optical density from the control groups (Figure 10H). Thus, the injury caused a decrease in the Pax2 expression (*p* < 0.05) in the pallial and subpallial areas that are not adjacent to the injury area.

## 3. Discussion

### 3.1. GS Localization in the Telencephalon of Intact Juvenile Chum Salmon and Those Exposed to Injury

The study of localization of GS (aNSCs/NPCs marker) in the telencephalon of juvenile intact and injury-exposed fish showed the characteristic significant structural rearrangements and the appearance of RG cells in the pallial and subpallial parts.

In the intact brain, GS labels a heterogeneous population of NPCs localized in PVZ, as well as a limited number of NPCs in parenchyma. In the medial pallial zone (DM) and all subpallial zones, GS labeling was detected in the granules located in PVZ (Appendix A). In the pallium, GS most frequently labeled NE cells in PVZ, which had an undifferentiated phenotype and lacked processes (Figure 5A–D). Similar GS^+^ cells of the NE type have also been found in the dorsolateral pallium of zebrafish [3]. However, studies show that neuronal precursors in the zebrafish telencephalon are characterized by spatial heterogeneity [4]. Previous studies on zebrafish revealed a granular pattern of expression for glial cell-derived neurotrophic factor (GDNF), which is believed to be involved in neurogenesis and neuronal regeneration [30]. An analysis of our data concerning the morphometric and densitometric parameters of GS^+^ cells in the pallium of juvenile chum salmon confirmed the heterogeneous pattern of labeled NPCs. In our studies, GS labeling was found in all pallial zones of the telencephalon. The pattern of labeling made it possible to distinguish both rather extensive neurogenic regions containing GS^+^ cells and single clusters, morphologically corresponding to local neurogenic niches of the constitutive type.

To characterize the rate of NPC proliferation and the fate of new neurons, the pallial and subpallial neurogenic zones of fishes are currently considered from the comparative aspect of the subgranular (SGZ) and subventricular (SVZ) neurogenic zones of mammals [31]. Thus, in studies on zebrafish, a low rate of proliferation and generation of neurons in the pallial region has been recorded, which allows comparison of this region with SGZ of the mammalian hippocampus [3]. In addition, short migration distances of pallial cells from PVZ to the parenchyma were identified in zebrafish [31]. Studies on the juvenile masu salmon pallium have shown that, after labeling of proliferating cell nuclear antigen (PCNA), more PCNA^+^ cells are concentrated in the medial zone than in the dorsal and lateral zones [32]. According to the results of investigation of cells’ proliferative activity after TBI using 5-bromo-2′-deoxyurenedine (BrdU) labeling in masu salmon, among the proliferating population of pallium cells, Dd cells are mainly in the S-phase, as well as at other stages of the mitotic cycle, including the state of migration to Dm. Both PCNA labeling and administration of BrdU have detected immunopositive cells and nuclei in Dm [6,32].

In zebrafish pallium, an alternating pattern of dividing PCNA^+^ cells and resting cells (PCNA–) has been revealed [1,33]. About two-thirds of PCNA^+^ cells are phenotyped as radial glia, both in terms of morphological traits and in positivity for radial glia markers such as glial fibrillar acidic protein (GFAP), vimentin, S100B protein, Notch and HER4, glutamine synthetase, specific astrocyte glutamate transporter, and fatty acid binding protein [31,33,34,35]. A third of PCNA^+^ cells are characterized as differentiating neuroblasts as they express the neural protein polysialylated cell adhesion molecule (PSA-NCAM) [1,33,36,37]. In the pallium of intact juvenile masu salmon, radial glial cells have not been detected, but proliferation markers (PCNA and BrdU), GFAP, and vimentin have been found in NE cells [6,32]. The data of the present study also indicate the presence of GS^+^ cells of the NE type exclusively in the constitutive neurogenic niches (CNN) of the intact juvenile chum salmon pallium.

In the subpallial area of juvenile chum salmon, the distribution of GS^+^ cells is more consistent with the localization of CNN. Thus, we can conclude that in the telencephalon of intact chum salmon GS a population of neuronal progenitor cells is labeled with an NE phenotype.

The subpallial zone of the fish telencephalon is currently considered as an analogue of the subventricular zone (SVZ) of mammals [31,33,35,38]. The subpallial area of zebrafish is generally characterized by a higher level of proliferation as compared to the pallial area and the presence of nestin-expressing proliferating cells [31,38,39]. These cells exhibit an NE phenotype, as well as interkinetic nuclear migration, expression of apical markers in *zonula occludens* (ZO1), and parvalbumin (PAR) PAR proteins [40]. The granular pattern of GS distribution, not only in PVZ, but also in SVZ, as well as the different intensities of immunolabeling, indicate a wide and heterogeneous structure of the subpallial neuronal zone. In the subpallial region of zebrafish, resting cells have not been identified; in this area, neurons are generated, migrate to the olfactory bulb, and differentiate into gamma-aminobutyric acid (GABA+) and tyrosine hydroxylase-ergic (TH+) interneurons in the granular and periglomerular layer [31].

We observed significant changes in GS localization in the injured telencephalon of juvenile chum salmon on day 3 post-injury (Figure 6A). We also detected the emergence of an additional type of GS^+^ neuronal precursors with a radial glia phenotype that is absent in intact animals (Figure 6, Figure 7, and Figure 8, Appendix A). The traumatic damage to the telencephalon led to a significant increase in the number of GS^+^ cells in PVZ, which was significantly higher than the number of immunopositive cells in the intact animals (Figure 8G). As a result of the injury, the intensity of GS labeling was also divided into two similar groups (intense and moderate); however, the level of OD after the injury significantly (*p* < 0.05) increased in both groups vs. the control (Figure 6G).

The heterogeneous population of GS^+^ cells after the injury included two types of neuronal progenitors. The first, NE, was previously present in CNN and was presumably involved in the process of RNN reactivation (Appendix A). Another type of neuronal progenitor, absent in intact animals, was a heterogeneous population of RG detected in RNN (Figure 6C, Appendix A). The distribution density of reactive RG in the pallial region of the telencephalon after injury was assessed in DD, DM, and DL, and, in our opinion, was in accordance with the concentration of neuronal precursors. The results of the comparative analysis showed that in the pallium the maximal density of RG distribution was in the DD (Figure 6H), and the minimal was in the DL (Figure 7H). The results of a comparative analysis indicated the predominance of RG in the pallial region (DD, DM, and DL) compared with the subpallial one (VD, VV).

Along with the appearance of reactive GS^+^ glial-type aNSCs, significant patterns of cell migration were revealed in the pallium and subpallium of juvenile chum salmon (Figure 6B). The trajectories of cell movement included radial (along RG fibers) and tangential patterns (Figure 6D,F), as well as the formation of numerous RNNs of different sizes, localizations, and lengths in the pallium (Figure 6E). In DM, the patterns of post-traumatic migration of GS cells formed a pseudo-stratified structure in SVZ and PZ (Figure 7A); in PVZ, numerous small RNNs, including neuroepithelial and radial glial GS^+^ cell populations, dominated. DL included patterns of tangential migration of GS^+^ neuroepithelial cells (Figure 7F), as well as areas containing the most extensive RG fibers (Figure 7F). The proportion of GS^+^ cells in the pallial and subpallial regions of the injured telencephalon increased significantly compared to GS– cells (Figure 8H).

According to published data on zebrafish brain, there are rapidly and slowly proliferating RG populations whose cells divide by asymmetric mitosis. As a result of such division, some of the daughter cells become devoid of process, and the others acquire a process which can subsequently be retracted by somal translocation. In our studies, after a traumatic injury to the telencephalon of chum salmon, GS^+^ cells of both types were found in RNN. Thus, as a result of the injury to the telencephalon, we observed patterns of CNN reactivation with the emergence of an additional type of radial–glial precursors labeled with GS (Figure 6 and Figure 7). After the injury, along with GS-labeled radially directed fibers, we identified an additional type of GS^+^ fibers in DL that were directed tangentially and propagating at the level of PVZ and SVZ (Figure 7F). Such fibers were absent in the control animals (Figure 5D).

Studies on zebrafish exposed to experimentally induced injury report about an intense regeneration processes [20,33,36]. The protocol used in our studies is similar to the previously published protocol on zebrafish [20]: a needle is injected into one of the two hemispheres of the telencephalon. The results have shown that the brain injury in *O. keta* causes cascading events characterized by edema, death of damaged neurons, glial hypertrophy, and an acute inflammatory response. The zebrafish studies have shown that the intensity of the inflammatory response reaches its peak on day 4 post-injury [33]. We observed intense cell proliferation both in neurogenic areas and in the parenchyma at 3 days after the damage to the telencephalon, which is consistent with the data for zebrafish [20]. During this period, zebrafish showed a high expression of mitotic markers such as PCNA in the SVZ zone [20] and in the dorsal region with GFAP+/S100B+ her4.1+ radial glial cells [33,36]. In studies on juvenile masu salmon *O. masou* at 1 week post-injury, the reactivation of GFAP+ and vimentin+ RG was revealed in DD and DL, and single, intensely small GFAP+ and vimentin+ cells were detected in the parenchyma. These elements emerged de novo as a result of the activation of resident glial-type aNSCs and their subsequent slow proliferation in response to the injury. This is expressed to the greatest extent in DL of juvenile masu salmon, and to the least extent in DM [6].

In the ventral zone, there were also changes in the localization and morphological structure of cells caused by the injury. As in the dorsal region, a population of cells with the RG phenotype appeared in the ventral and dorsal nuclei; however, the number of such cells was significantly smaller than in the dorsal region. No RG cells were found in the VL zone, but the intensity of GS immunolabeling increased as compared to the control (Figure 8C).

Thus, as a result of the traumatic injury to the telencephalon, the level of GS expression in various types of progenitors increased multifold (Figure 6G). Our data agree with the results of studies on the damaged cerebellum of *Apteronotus albifrons* [41], in which a significant increase in the GS content in the brain was recorded after the injury. According to Zupank, the increased production of GS is a response to an increase in the amount of extracellular glutamate induced by damage. In the fish brain, the production of GS that converts toxic glutamate into neutral glutamine, on the contrary, increases in response to traumatic injury [2,12]. In our studies we observed similar effects associated with the increased GS production among PVZ cells containing neuronal precursors. We suggest that GS is not only a marker of NE and RG precursors that are activated by injury but can also be produced by some cells located in the parenchyma in order to neutralize glutamate. In this sense, GS is considered by us as a neuroprotective, regenerative-associated factor that promotes effective reparative neurogenesis after injury.

### 3.2. Localization of CBS in the Intact and Injured Telencephalon of Juvenile Chum Salmon

Despite regional differences, the common features of CBS localization were identified as follows: extended zones of distribution of intensely labeled cells alternating with areas that lack immunopositivity, which corresponds to the patterns of GS distribution of NE cells in the intact telencephalon. Previous studies of the localization of PCNA in the telencephalon of juvenile masu salmon *O. masou* showed that the patterns of constitutive proliferation in the territory of the pallium are represented by local small-sized constitutive niches that occupy PVZ [32]. The patterns of neurogenesis revealed using the marker of neuronal differentiation HuCD indicate that in intact juvenile masu salmon the cells labeled with HuCD form rather extensive groups alternating with immunonegative regions [32]. In the absence of a differentiated cellular structure in the telencephalon of juveniles, H_2_S-producing cells in the brain parenchyma release hydrogen sulfide into the intercellular space, in which young neurons are differentiated, thereby creating a special microenvironment that promotes neuronal differentiation. Thus, hydrogen sulfide secreted by cells can be considered as a factor involved in constitutive neurogenesis. This is confirmed by the results of studies on mammals in which a high level of CBS expression was recorded from the hippocampus, which is characterized by neurogenesis in the adult state [42]. According to these data, a high level of CBS is required for early maturation and growth of neuronal networks, which also confirms the role of hydrogen sulfide in neuronal differentiation [42].

The results of IHC labeling with CBS in the telencephalon of juvenile chum salmon showed the presence of intensely and moderately labeled cells in the pallial and subpallial regions of the telencephalon, which is consistent with the previously obtained data on trout [17]. Intensively labeled cells in *O. keta*, as in trout, were located in the superficial periventricular and subventricular layers of the pallium. In deep pallial parenchymal areas, the number of intensely labeled cells in *O. keta*, like in trout, was increased. In addition to the cellular localization of CBS, immunopositive granules were also detected in the pallium (DD and DM) and in all areas of the subpallium of juvenile chum salmon (Appendix A
Appendix A). The presence of an H_2_S-producing enzyme in brain cells is associated with the neurochemical signaling and, in particular, with the activation of NMDA receptors [14,43]. The activation of neurons in the brain causes neurotransmitters, including glutamate, to be released, thus, activating NMDA receptors, which, in turn, leads to an increase in astrocytic intracellular calcium [43,44]. Therefore, the presence of two levels of CBS activity in cells and granules of the juvenile chum salmon telencephalon indicates mediator–modulatory intercellular interactions, which is consistent with the previously obtained data on fish [16,17,45].

As a result of injury to the *O. keta* telencephalon, the number of H_2_S-producing cells increases sharply, which leads to an increase in the total production of hydrogen sulfide in the brain. Taking into account the fact that the rates of such processes as proliferation, cell migration, and neuronal differentiation in PVZ after traumatic injury increase multifold, we may consider hydrogen sulfide as a factor contributing to reparative processes, which is consistent with the previously obtained results of a study of unilateral eye injury in trout [17,18] and cerebellum of masu salmon [16]. Due to the high efficiency of the regenerative process in the fish brain, it can be concluded with confidence that the increased production of H_2_S in the NE cells of the juvenile chum salmon brain after the injury is directly related to the processes of reparative neurogenesis. The results of the study on the telencephalon of juvenile *O. keta* are consistent with the previously obtained results on *O. masou* which also revealed the effects of increased H_2_S production in the cerebellum 3 days after injury [16]. Unlike trout [18], no RG-type CBS^+^ NPCs were detected in the pallium of *O. keta*. We associate this with the later adult stage of ontogenesis of trout, whose pallium contains a greater number of neuronal precursors of the glial type (aNSC) with the RG morphology.

Experimental results have shown that the toxicity of glutamate during traumatic injury is attenuated by the action of H_2_S on voltage-gated ATP-sensitive potassium channels K_ATP_ and fibrosis transmembrane conductance regulator CFTR Cl-channels [46] and activation of glutamate GLT1 transporter [47]. However, there is currently no consensus on the dual role of H_2_S in glutamate toxicity. The neuroblasts formed in the matrix periventricular zones of the telencephalon in juvenile chum salmon are immature cellular forms that can express an incomplete set of glutamate NMDA receptors, migrating from the periventricular to the subventricular layers of the brain. In such undifferentiated cells, the cascade processes leading to apoptosis in mature neurons can be inhibited and, as a result, immature cells can avoid death and be successfully involved in the processes of neuronal regeneration.

The insufficient supply of oxygen and glucose to the brain tissue in case of impaired blood circulation after a brain injury leads to a decrease in oxidative phosphorylation in neurons. As a result of reoxygenation, the amount of reactive oxygen species increases, which disrupts the functions of neuronal mitochondria and, as a result, further reduces the ATP production [48]. Loss of ATP leads to an imbalance of ionic homeostasis in the telencephalon cells due to the failure of ATPases or carriers of ATP-dependent ions [22] which regulate the influx of calcium and sodium. The resulting changes induce the outflow of potassium due to the subsequent depletion of ATP and calcium accumulation [49,50]. An increase in intracellular calcium leads to the production of glutamate, which increases calcium overload and activates calcium-dependent lipases and proteases [50]. The significant shifts in ionic homeostasis cause an increase in the production of reactive oxygen species, the opening of transition pores of mitochondrial permeability, inflammation, and neuronal death [22,51]. In intact mammals, astrocytes surrounding neurons absorb extracellular glutamate and protect neurons from excitotoxicity [49,52]. However, the data obtained on fishes after TBI differ significantly from the results of a biochemical analysis with determination of H_2_S in the mammalian brain [53], where a decrease in H_2_S production was observed after a traumatic injury.

Reparative neurogenesis in the mammalian brain is significantly hampered by the processes of secondary inflammation arising as a result of the toxic effects of glutamate and high excitotoxicity [48,49,50,51], while in the fish brain, on the contrary, no similar effects develop. In intact mammals, astrocytes surrounding neurons absorb extracellular glutamate and protect neurons from excitotoxicity [50,51,52]. However, the data obtained on fishes after brain injury differ significantly from the results of a biochemical analysis with determination of H_2_S in the mammalian brain [53], where a decrease in H_2_S production was observed after a traumatic injury. We assume that the increased production of H_2_S and GS in the telencephalon after the traumatic injury facilitates the reparative process. The increase in glutamate excitotoxicity after a telencephalic injury in *O. keta* can be neutralized by H_2_S, which reduces the toxic effects of glutamate. The H_2_S produced in the telencephalon can be considered as a proneurogenic and neuroprotective factor [44]. Thus, the increase in H_2_S production in the brain after injury should be explained from the point of view of maintaining cerebrovascular homeostasis, which implies antiapoptotic, anti-inflammatory, and antioxidant effects that reduce the level of secondary neuronal damage resulting from oxidative stress.

### 3.3. Molecular Basis of Neuronal Regeneration. Localization of Pax2 in the Intact and Injured Telencephalon of O. Keta

As is known, Pax2 expression is regulated by the Shh signaling pathway [54,55]. However, it was found that during optic stem development Pax2 expression is regulated by the relationship between the Shh and bone morphogenetic protein (BMP) cell signaling pathways [54]. Segal and co-authors [55] have shown that the secondary messengers of the BMP and Shh pathways are expressed in the developing optic stem and together inhibit Pax2-suppressive effects. Disturbances in Shh signaling in the ventral midline cause the Pax2 expression to decrease and the Pax6 expression to increase in the retina and in the region of the presumptive optic stem [54,56]. The mutual inhibition of Pax2 and Pax6 in the ventral region of the developing eye distinguishes progenitor cells into gliogenic (Pax2-positive) and neurogenic (Pax6-positive) populations [57]. When considering these results, it is possible to assume that Pax2 provides the glial phenotype of cells and suppresses the neurogenic potential of NPCs in the chum salmon telencephalon. Our results are consistent with the data on immunofluorescent labeling that have shown the Pax2 distribution in the retina of adult guinea pigs, chickens, and mice [58].

The study on juvenile chum salmon has shown that Pax2 labels the population of neuroepithelial cells in different regions of the telencephalon. The expression of Pax2 in juvenile chum salmon was detected both in cell cytoplasm and in nuclei. Thus, in intact juvenile chum salmon the Pax2 labels a limited population of neuroepithelial cells and nuclei of unidentified cells, which, as we understand, participate in the processes of constitutive neurogenesis. Taking into account some data on the involvement of the PAX genes family in the construction of the spatial structure of CNS during embryogenesis and postembryonic development [59], we suggest that the Pax2 expression in cell nuclei in the deep layers of parenchymal tissue reflects the brain patterning in growing *O. keta*. This assumption is consistent with the data on the involvement of Pax2 in the process of regionalization of CNS in the embryonic and postembryonic periods of vertebrate development [59].

After the brain injury, we found a population of tangentially migrating, intensely labeled Pax2 cells in the surface and basal parts of PVZ. This pattern of localization of Pax2^+^ cells was typical for almost all regions of the telencephalon, which indicates the generalized expression of this TF. We identified the patterns of Pax2^+^ migrating cells in the area of injury, which is the final point of cell migration. Thus, after the injury to the telencephalon, we found that Pax2 expressing cells retain the function of spatial patterning of CNS, but its vector changes: in particular, instead of the population of resident NE cells, an intense Pax2 expression occurs in the population of migrating cells involved in the reparative reorganization of CNS. Nevertheless, the quantitative analysis showed a significant decrease in the number of Pax2^+^ cells in almost all areas of the pallial (DM, DL, *p* < 0.05) and subpallial (VL, *p* < 0.05) zones of the juvenile chum salmon telencephalon (Figure 10G). The Pax2 expression in cells remains at the level of intact expression or slightly decreases (Figure 10H). Our data can be interpreted taking into account the previously established mutually inhibiting properties of Pax2 and Pax6 expression [57,58]. The previously studied patterns of Pax6 expression in the trout telencephalon after a unilateral damage to the optic nerve showed a significant increase in the Pax6 expression in precursors of the neuroepithelial and glial types [26]. After the optic nerve injury, an intense Pax6 labeling was detected in RNN located within the pallial proliferative zones (DD). In this region of the trout telencephalon, we did not find Pax6+ RG; however, the immunolabeling of periventricular cells in the pallial zone indicates a significant increase in the production of neuroepithelial Pax6+ cells in RNN with the nuclear localization of Pax6. In DL, Pax6+ RNN of periventricular and subventricular localizations, as well as RG fibers, along which Pax6+ and Pax6– cells migrated, were identified after the optic nerve injury. Similar patterns of localization of Pax6+ RNN were also typical for subpallial areas [26]. Immunoblotting data also indicate that the amount of Pax6 in trout after the optic nerve injury significantly increases compared to the level of expression in intact animals [26,27].

Several splicing variants of Pax2 were identified, although their functional significance is not yet understood. In *Xenopus*, the Pax2 expression has at least nine splicing variants [60], and five variants have been recorded from humans [61,62]. We suggest that the different combinations of binding partners, phosphorylation states, DNA/promoter methylation, and splice variants determine whether the reciprocal inhibition of Pax6 and Pax2 occurs. A study of the co-expression of Pax2 and Pax6 in the chicken retina has shown, however, that the Pax2 isoforms expressed in Müller glia cells are not capable of inhibiting the Pax6 expression [58].

In our opinion, the patterns of Pax2 expression in the juvenile chum salmon telencephalon on day 3 post-injury correspond to neuroepithelial NPCs, both resident and reactivated by injury, and are reduced compared to those in intact animals. We suggest that the decrease in Pax2 expression is associated with the inhibitory effect of Pax6, whose expression is, on the contrary, significantly increased. Further studies are required to clarify in detail the mechanism of this process.

## 4. Material and Methods

### 4.1. Experimental Animals

The material used in the work was 30 individuals of juvenile *Oncorhynchus keta* aged 12–18 months. The body weight of the animals was 31–69 g; the body length, 15–22 cm. The material was sampled at the Ryazanovka experimental fish hatchery in 2019. The animals were kept in a tank with aerated fresh water at a temperature of 15–16 °C and fed once a day. The light/dark cycle was set at 14/10 h. The content of dissolved oxygen in water was 7–10 mg/dm^3^, which corresponds to normal saturation. All experimental manipulations with animals were carried out in accordance with the rules regulated by the charter of the Zhirmunsky National Scientific Center of Marine Biology (NSCMB) FEB RAS, the Resource Center of the NSCMB FEB RAS, and the Ethics Committee, which regulates the humane treatment of experimental animals (approval no. 2-170920 from Meeting No. 1 of the Commission on the biomedical ethics of NSCMB FEB RAS, 17 September 2020).

### 4.2. Experimental Injury to the Telencephalon by the Kishimoto’s Method 

Animals were anesthetized in a cuvette containing 0.01% ethyl-3-aminobenzoate methanesulfonate (MS222) (Sigma, St. Louis, MO, USA, Cat. # WXBC9102V) for 5 min at room temperature. Using a sterile needle, a mechanical injury was applied to the region of the dorsolateral quadrant of the right hemisphere of the telencephalon to a depth of 1 mm. Immediately after the injury, the animals were released back into the tank for recovery and further monitoring.

After the damaging effect in the telencephalon region, changes in the locomotor and behavioral activity in fish in the experimental group were monitored for 1 h. There were no significant changes in the locomotor activity in animals with the telencephalon injury compared with the control group. A small hematoma measuring 1–2 mm was clearly visible in the area of injury.

### 4.3. Sample Preparation

At three days post-injury, all the experimental fish were deeply anesthetized in a 0.01% solution of MS 222 (Sigma, St. Louis, MO, USA) for 5 min to collect telencephalon samples. Animals were perfused with 4% paraformaldehyde solution (PFA, BioChemica, Cambridge, MA, USA; Cat. No A3813.1000; lot 31000997) prepared in 0.1 M phosphate buffered saline (PBS, pH 7.2) (Tocris Bioscience, Minneapolis, MN, USA; Catalog No 5564, Batch No.: 5). After prefixation, the brain was removed from the cranial cavity and fixed in the same solution for 2 h at a temperature of 4 °C. Then, it was washed in a 30% sucrose solution at 4 °C for two days, with five changes of the solution. Serial frontal sections (50 μm thick) of the brain were cut on a freezing microtome (Cryo-star HM 560 MV, Walldorf, Germany).

### 4.4. Immunohistochemical Detection of CBS, Pax2, and GS

To study the localization of GS, CBS, and TF Pax2 in the telencephalon of juvenile chum salmon, immunoperoxidase labeling on frozen free-floating sections was used. Prior to immunostaining of the fish telencephalon tissue, the activity of endogenous peroxidase and nonspecific staining (background) was blocked. The endogenous peroxidase activity was blocked by incubation with 1% hydrogen peroxide for 20 min at room temperature. To eliminate nonspecific staining, the brain sections were incubated with non-immune horse serum. After incubation and washing with 0.1 M PBS, the frozen 50-μm brain sections were incubated in situ with primary rabbit anti-Pax2 polyclonal antibodies (Biolegend, San Diego, CA, USA; Catalog No. 901001; 1:300), rabbit anti-CBS polyclonal antibodies (GeneTex, Irvine, CA, USA; Catalog No. GTX124346; 1:300), and mouse monoclonal antibodies against GS (Abcam, Cambridge, UK; Catalog No. ab64613; 1:300) at 4 °C for 48 h. The anti-rabbit streptavidin–biotin imaging system (HRP conjugated Anti-Rabbit IgG SABC Kit; Boster Biological Technology, Pleasanton, CA, USA; Catalog No. SA1022) and the avidin–biotin (ABC) complex (Vectastain Elite ABC kit; Vector Laboratories, San Francisco, CA, USA; Catalog No. PK-6100) were used for visualization of immunohistochemical (IHC) labeling products of polyclonal rabbit and monoclonal mouse antibodies, respectively. To identify the reaction products in both cases, a red substrate was used (VIP Substrate Kit; Vector Labs, Burlingame, CA, USA; Catalog No. SK-4600). Then, to identify immunonegative cells, the preparations were counterstained with a 0.1% methyl green (MG) solution (Bioenno, Lifescience, Santa Ana, CA, USA, Cat # 003027). The color development was monitored under a microscope, washed with distilled water for 10 s, and then differentiated for 1–2 min in a 70% alcohol solution and then for 10 s in 96% ethanol. The preparations were dehydrated by a standard technique: they were kept in two changes of xylene, 15 min for each, and then placed under cover slips in the Bio-Optica medium (Milan, Italy).

To assess the specificity of the immunohistochemical (IHC) reaction, a negative control method was used. The brain sections were incubated with a 1% solution of non-immune horse serum instead of primary antibodies for 1 day and processed as sections with primary antibodies. In all the control experiments, there was no immunopositive reaction.

### 4.5. Microscopy

To visualize the IHC reaction in the telencephalon cells and conduct morphological analysis, we used a motorized microscope Axiovert 200 M equipped with a fluorescence ApoTome module and an attachment for improved contrast (Carl Zeiss, Oberkochen, Germany). The material was analyzed using the AxioVision software. Measurements were performed at 100×, 200×, 400× magnifications and in several randomly selected fields of view for each study area. The number of labeled cells in the field of view was counted at a magnification of 200×. Micrographs of the mounts were taken and analysis of the material was carried out using the Axio Vision software. The material was processed using the Axioimager program and the Corel Photo-Paint 14 graphics editor.

### 4.6. Densitometric Analysis

The optical density (OD) of IHC labeling products in neuronal bodies and immunopositive granules was measured and the subsequent morphometric analysis of the parameters of cell bodies (the greater and smaller diameters of the neurons’ soma) was carried out using the software of the Axiovert 200 M microscope (Carl Zeiss, Oberkochen, Germany). For this, OD was measured at several sites of immunolabeled structures in a mount. Then, using the Wizard software, a standard OD assessment was carried out for intensely and moderately labeled and immunonegative cells. The averaged value of OD for each type of cells was subtracted from the mean value of OD of the background and, thus, the actual value of relative units of optical density (UOD) was obtained.

### 4.7. Stereological Method in the Study of Quantitative Parameters of the Telencephalon

To obtain reliable quantitative characteristics of various regions of the telencephalon of juvenile *O. keta*, as a volumetric object in space, we used the stereological method of calculating the data obtained during microscopic analysis. For reliable spatial reconstruction for the ICH study, we used every 3 sections of the telencephalon. The stereological method enables, with the reliability determined by the objectives of the study and controlled by varying the parameters of the study design, the revealing of the morphometric characteristics of the object under study on the material of a limited number of sections. In this case, the systematic error (bias) is proven to be avoided, and the measurement error is controllable and directly depends on the sampling frequency: the more slices, the higher the accuracy. This is achieved through the use of an appropriate mathematical apparatus and adherence to sampling rules, in particular, a systematic random sampling.

When working with sections of the telencephalon, we selected the area under study, after which we took into account the morphometric parameters of interest. After receiving data from all selected slices, we performed calculations that allowed us to proceed to the description of the volumetric object. In particular, data were obtained on parameters such as the number of immunopositive cells within the dorsal, medial, lateral and central zones of the pallium and dorsal, ventral and lateral zones of subpallium, the density of distribution of immunopositive cells in constitutive and reactive neurogenic niches, and the number of immunopositive cells, nuclei and granules in the periventricular, subventricular and parenchymal zones of all parts of the telencephalon.

### 4.8. Statistical Analysis

Prior to the experiments, we performed a statistical analysis based on the variations in the measured parameters in our previous research [6] and determined that we needed a group of at least 4 animals to achieve the statistical confidence at 95%. To make sure that we reach a group size of 4 and, at the same time, reduce the use of animals to a minimum, we aimed for a total of 5 animals per experimental group.

The quantitative assessment of cells was carried out on a separately selected viewfield at a magnification of 10× of the objective and 20× of the eyepiece. All data analysis was performed using a blind test to reduce experimenter’s bias. The absolute number of cells of each type was counted. All quantitative results in the present study were analyzed with the SPSS software (version 16.0; SPSS Inc., Chicago, IL, USA), and expressed as the mean ± standard deviation of the mean (M ± SD), and differences were considered statistically significant at *p* < 0.05. The quantitative processing of morphometric data of IHC labeling was performed using the Statistica 12 and Microsoft Excel 2010 and STATA software packages (StataCorp. 2012, Stata Statistical Software: Release 12. College Station, TX: StataCorp, LP, College Station, TX, USA). All variables measured in the groups were compared using one-way analysis of variance (ANOVA) followed by the Student–Newman–Keuls post hoc test. Comparison of the number of immunopositive cells between several neuroanatomical areas of pallium and subpallium of the intact or injured telencephalon (presented as groups) was performed using one-way analysis of variance (ANOVA) with Bonferroni’s correction. Values at *p* ≤ 0.05 and *p* ≤ 0.01 were considered statistically significant.

## 5. Conclusions

In the intact brain of juvenile chum salmon, GS labels a heterogeneous population of neuroepithelial-like cells localized in the PVZ, as well as a limited number of neuroepithelial type NPCs in the parenchyma. In the medial zone of the pallium (DM) and all subpallial zones, GS labeling was detected in extracellular granules located in the PVZ. In the subpallial area, the distribution of GS^+^ cells corresponds to the localization of CNN. On day 3 post-injury, significant changes in the GS localization were revealed in the telencephalon such as, in particular, the appearance of an additional type of GS^+^ neuronal precursors with an RG phenotype that are absent in intact juvenile chum salmon. The results of a comparative analysis indicate the predominance of RG in the pallial regions (DD, DM, and DL) in the post-traumatic period as compared to the subpallial regions (VD, VV). The maximum distribution density of RG in the pallium was found in DD and the minimum one in DL. After the traumatic injury to the chum salmon telencephalon, the patterns of CNN reactivation were identified; GS^+^ cells of both the neuroepithelial and RG types were found in the RNN.

In the dorsal and ventral parts of the juvenile chum salmon telencephalon, CBS labels a heterogeneous population of periventricular and parenchymal NE cells. In intact fish, the following common features of CBS localization were revealed: extended zones of distribution of intensely labeled cells alternating with areas devoid of immunopositivity, which corresponds to the patterns of GS distribution over NE cells in the intact telencephalon. After the damage to the telencephalon, the number of CBS-labeled cells of the NE type in the PVZ of the pallium and subpallium increased significantly compared to that in the control animals. The quantitative assessment of CBS^+^ cells in all areas of the pallium and subpallium at 3 days after the telencephalon injury showed a significant increase in the number of CBS^+^ cells in all areas of the pallium (DD, DM, DL) (*p* < 0.05) and the ventral area of the subpallium (BB) (*p* < 0.05), as well as in VL (*p* < 0.01). Thus, as a result of the telencephalon injury, the number of H_2_S-producing cells increased sharply, which caused an increase in the total H_2_S production in the juvenile chum salmon brain. Based on the fact that the processes of proliferation, cell migration, and neuronal differentiation in the PVZ intensify after a traumatic injury, we may consider H_2_S as a proneurogenic and neuroprotective factor contributing to the maintenance of cerebrovascular homeostasis and having the anti-inflammatory and antioxidant effects that reduce the level of secondary neuronal damage resulting from oxidative stress.

Pax2 labels the population of NE cells in different regions of the juvenile chum salmon telencephalon. The Pax2 expression was detected both in cell cytoplasm and in nuclei. Thus, in intact fish, the Pax2 protein product labels the gliogenic population of cells and nuclei of unknown functional specialization, which, as we believe, is involved in the processes of constitutive neurogenesis. After the brain injury, a population of Pax2^+^ cells migrating tangentially towards the injury area was identified in the PVZ. Thus, as a result of injury to the telencephalon of juvenile chum salmon, the function of spatial CNS patterning in Pax2^+^ cells is retained, but its vector changes; instead of the population of resident NE cells, intense expression of Pax2 was found in the population of migrating cells involved in reparative reorganization of CNS. The quantitative analysis showed a decrease in the number of Pax2^+^ cells in almost all areas of the pallial (DM, DL; *p* < 0.05) and subpallial (VL; *p* < 0.05) parts of the telencephalon. We believe that the patterns of Pax2 expression in the juvenile chum salmon telencephalon on day 3 post-injury correspond to NPCs of both resident NE cells and reactivated trauma and are reduced as compared to those in intact fish. Further studies can clarify in detail the mechanism that we observed. However, we assume that the decrease in the Pax2 expression may be associated with the inhibitory effect of Pax6, with its expression, vice versa, increasing significantly after a traumatic injury.

The obtained data raise various questions to address in the further study of neurogenesis in the adult Pacific salmon brain as follows. How does the ratio of GS^+^ neuroepithelial and glial precursors change as a result of injury to the telencephalon of juvenile chum salmon? What role does H_2_S play in the initiation of the acute inflammatory response in the brain of juvenile *O. keta*? Why does the number of Pax2^+^ cells in the telencephalon decrease in the acute post-traumatic period, while the number of Pax6+ cells increases? The answers to these questions can significantly extend the current knowledge of salmon neurogenesis as a convenient model for studying brain plasticity, reparation and postembryonic development.

## Figures and Tables

**Figure 1 ijms-22-01279-f001:**
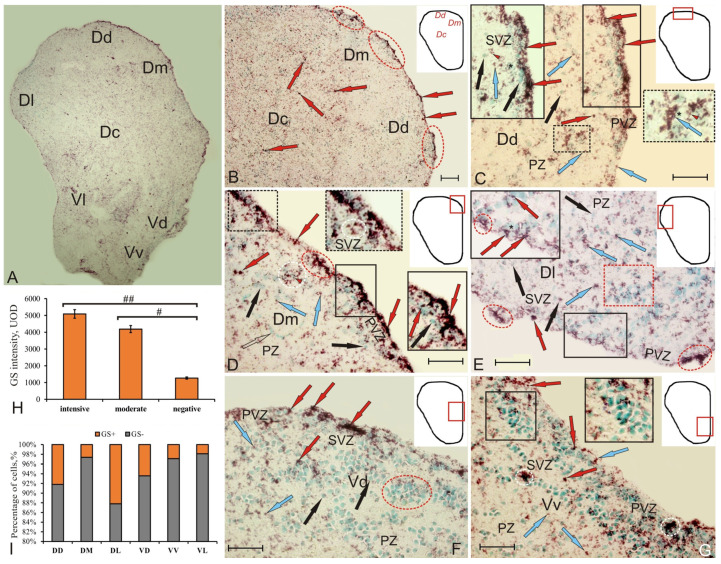
Glutamine synthetase (GS) in the pallial and subpallial regions of intact telencephalon in juvenile chum salmon, *Oncorhynchus keta*. (**A**) General view of the GS immunolocalization pattern in the telencephalon of intact juvenile chum salmon, Dd—dorsal, Dm—medial, Dc—central zones of pallium, Vd—dorsal, Vv—ventral, Vl—lateral zones of subpallium; (**B**) in the pallial area, the pictogram shows the zones of the dorsal telencephalon (pallium), ovals outline periventricular aggregations of GS^+^ cells, intensely labeled cells (red arrows); (**C**) dorsal pallial zone (Dd) at higher magnification, inset (in red rectangle) shows a fragment including periventricular zone (PVZ) and subventricular zone (SVZ); inset (in black rectangle) shows intensely labeled GS^+^ neuroepithelial cells (red arrows) located in the PVZ above the layer of immunonegative cells (black asterisk), in the SVZ there were (dotted inset) clusters of small moderately labeled cells (red triangular arrow), a moderately GS^+^ cell (blue arrow), GS–cell (black arrow), PZ—parenchymal zone; (**D**) the medial pallial zone (Dm) contains small GS^+^ cells (outlined with a white dotted line) in SVZ (black dotted inset), dense groups of small intensively labeled cells in PVZ of irregular shape (inset in a black rectangle), endothelial cells are shown with a transparent arrow; (**E**) lateral pallial zone (Dl) including small moderately labeled GS cells (in the red oval) and GS^+^ cytoplasm of neuroepithelial cells surrounding the GS nuclei in PVZ (inset in the black rectangle), the neurogenic niche of parenchymal localization is in the red rectangle; (**F**) the dorsal subpallial zone (Vd) contains intensely labeled groups of neuroepithelial cells (red arrows) and parenchymal neurogenic niches (in red oval); (**G**) the ventral subpallial zone (Vv) contains single (red arrows) or paired (blue arrows) moderately labeled cells in the apical part of PVZ, aggregations of intensely labeled granule-like subcellular elements (in white ovals), pseudo-unipolar GS^+^ cells occurred in SVZ and penetrate into the deeper layers of PZ (inset). Immunohistochemical labeling of glutamine synthetase. Scale bar: 100 µm. (**H**) Comparative densitometric analysis of labeling of GS^+^ cells; significant intergroup differences # (*p* < 0.05), ## (*p* < 0.01) between the groups of intensely, moderately labeled cells, and immunonegative cells (*n* = 5 in each group). One-way ANOVA. Intensity of GS labeling, units of optical density (UOD). (**I**) Percentage ratio between GS^+^ and GS–cells in all intact pallial and subpallial areas of juvenile chum salmon, *Oncorhynchus keta*.

**Figure 2 ijms-22-01279-f002:**
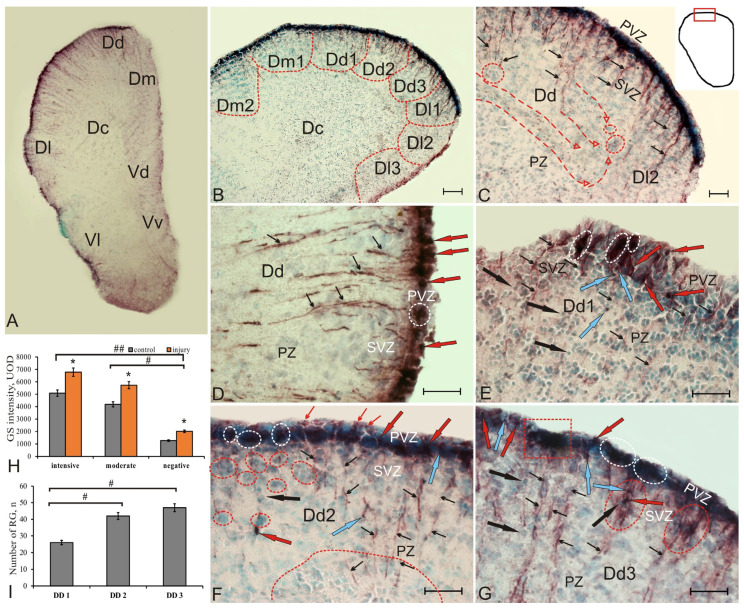
Glutamine synthetase in the pallial region of the telencephalon of juvenile chum salmon, *Oncorhynchus keta*, at 3 days after traumatic injury. (**A**) General view of the GS immunolocalization pattern in the damaged telencephalon of juvenile chum salmon, (**B**) pallial zone, the red dotted line indicates the boundaries of the subzones; (**C**) patterns of cell migration (directions are indicated by red dashed arrows) and distribution of radial glia (black arrows) in DD, reactive neurogenic niches of parenchymal localization are outlined by red ovals; (**D**) patterns of distribution of radial glia (RG) and reactive neurogenic niches (in white oval) in DD; (**E**) cellular composition of reactive neurogenic niches in DD1, small reactive intensely labeled GS clusters of cells (in white ovals), GS–cells are indicated by black arrows; (**F**) dense clusters of GS + RG in DD2, thin red arrows indicate the end-feets of the immunopositive RG in PVZ, the red dotted line outlines the reactive aggregations of neuroblasts in SVZ and the largest in PVZ; (**G**) patterns of RG distribution in DD3, single radial fibers (thin black arrows), along which immunonegative neuroblasts migrate (black arrows), combined with bundles of radial fibers (red ovals); large reactive neurogenic niches (RNN) (in a red rectangle). Immunohistochemical labeling of glutamine synthetase. Scale bars: (**B**) 100 µm, (**C**–**G**) 50 µm. (**H**) Comparative densitometric analysis of GS^+^ cell activity in intact animals and on day 3 post-injury; significant intergroup differences # (*p* < 0.05), ## (*p* < 0.01) between the groups of intensely, moderately labeled cells, and immunonegative cells (*n =* 5 in each group). Data are expressed as the mean ± SD (*n =* 5 in each group); one-way ANOVA followed by the Student–Newman–Keuls post hoc test was used to determine significant difference between control animals and those on day 3 post-injury, *p* < 0.05 vs. control group. Intensity of GS labeling, UOD; * *p* ≤ 0.05. (**I**) Comparative distribution of RG in DD1, DD2 and DD3, on day 3 post-injury. One-way ANOVA (*n =* 5 in each group, #—significant intergroup differences (*p* < 0.05).

**Figure 3 ijms-22-01279-f003:**
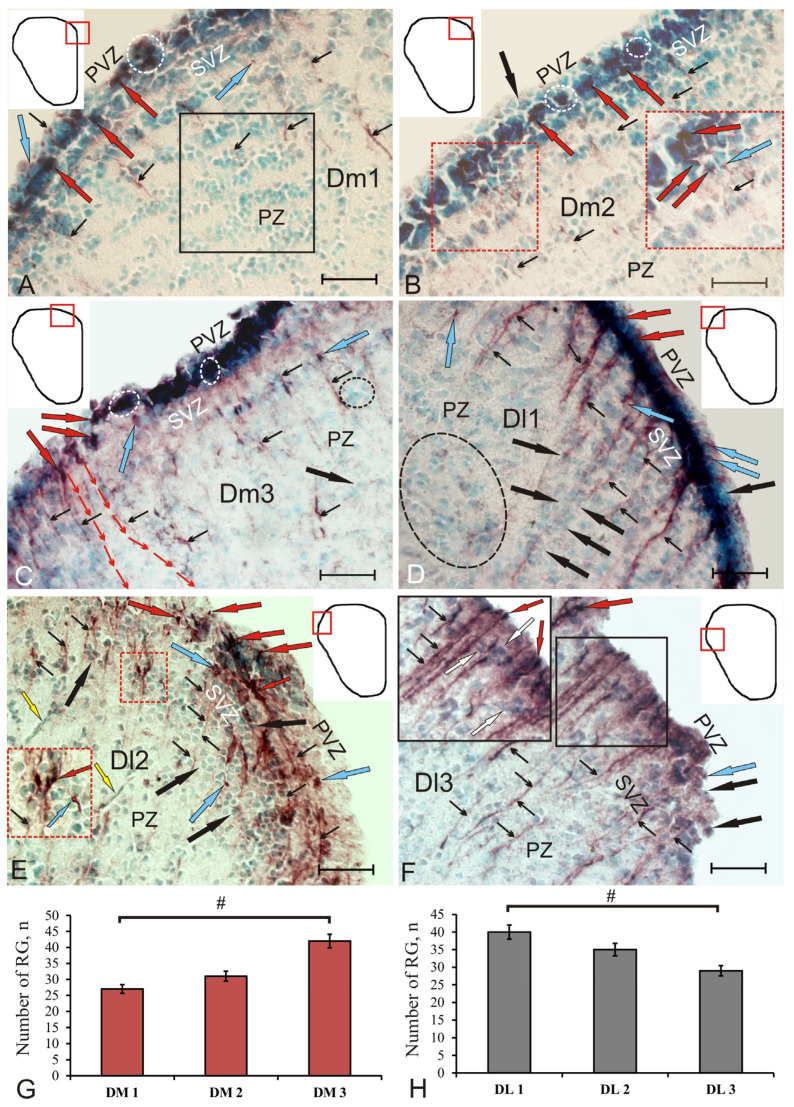
Glutamine synthetase in the medial (**A**–**C**) and lateral (**D**–**F**) pallial areas of the telencephalon of juvenile chum salmon, *Oncorhynchus keta*, at 3 days after the traumatic injury to the telencephalon. (**A**) Diffuse pattern of GS^+^ radial glia distribution in DM1, densely or moderately labeled GS^+^ neuroepithelial cells (blue arrows), a fragment of the pseudo-stratified structure of PZ with migrating neuroblasts and RG fibers is in the black rectangle; (**B**) reactive neurogenic niches containing GS^+^ neuroepithelial cells (dotted inset) in DM2; (**C**) patterns of neuroblasts’ migration in DM3 (the direction of migration is indicated by red thin arrows), RNN of the neuroepithelial type in PVZ (in white ovals), RNN of parenchymal localization (in the black oval); (**D**) the largest and longest RG fibers and bundles in DL1, with GS–neuroblasts migrating along RG (black arrows), aggregation of GS neuroblasts (in the black oval); (**E**) patterns of tangentially oriented GS^+^ fibers in DL2; along the RG fibers, extensive migration of a heterogeneous population of cells was observed (inset), yellow arrows indicate endothelial cells; (**F**) dense population of GS + RG in DL3, long outgrowths deeply spreading in PZ (inset). Immunohistochemical labeling of glutamine synthetase. Scale bar: 50 µm. (**G**) Comparative distribution of RG in DM1, DM2 and DM3, and in DL1, DL2 and DL3 on (**H**) day 3 post-injury. One-way ANOVA (*n =* 5 in each group, #—significant intergroup differences (*p* < 0.05).

**Figure 4 ijms-22-01279-f004:**
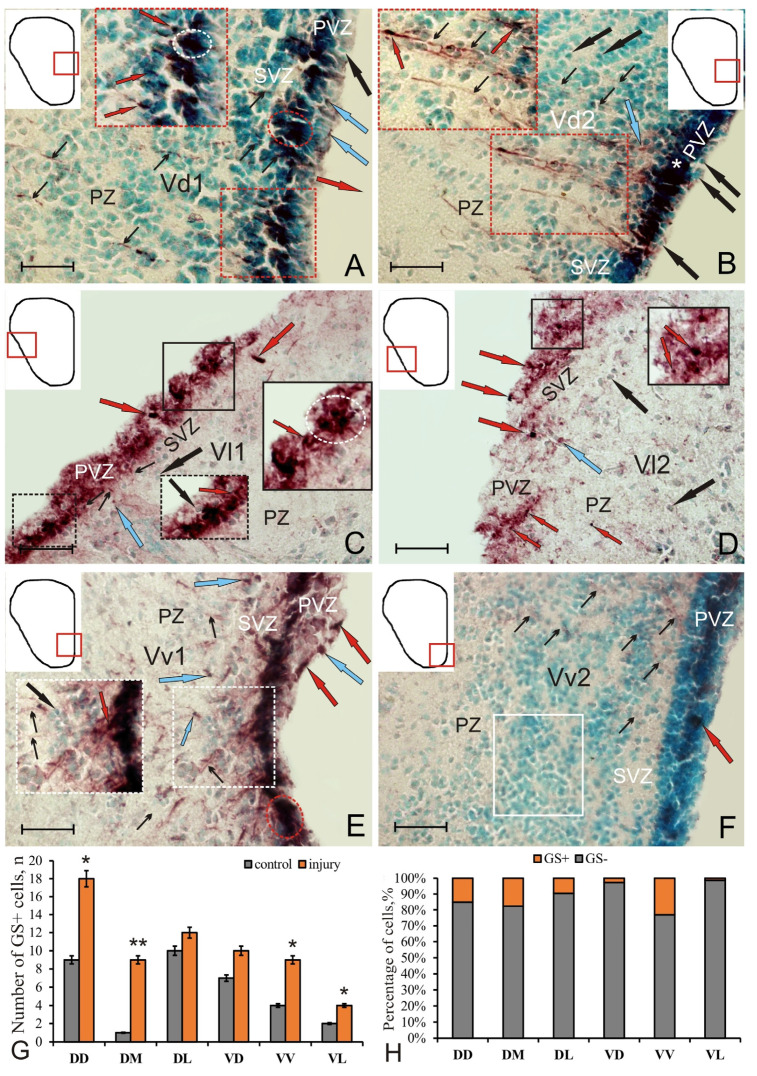
Glutamine synthetase in the subpallial areas of juvenile chum salmon, *Oncorhynchus keta*, at 3 days after the traumatic injury to the telencephalon. (**A**) In VD1, numerous neuroepithelial cell clusters were detected in PVZ and SVZ (inset), producing the RNN (in the red oval), and the number of RG fibers (black thin arrows) was reduced; (**B**) migrating GS^+^ cells were found in VD2 along GS^+^ RG fibers (red arrows, inset); in the PVZ, elongated GS^+^ cells (white asterisk) were revealed, above which a layer of GS–neuroepithelial cells was found (black arrows); (**C**) VL1 was dominated by aggregations of GS^+^ neuroepithelial cells (in the white oval) in the dorsal part of PVZ (black inset) and in the ventral part (dotted inset), and the number of RG (thin arrows) was limited; (**D**) single GS^+^ neuroepithelial cells (red arrow) were detected in VL2 in PVZ (inset), and RG fibers were not revealed; (**E**) in VV1, mixed RNNs were found (inset) containing large densely stained GS^+^ groups of neuroepithelial cells (red arrows), RG fibers (thin black arrows), and individual moderately labeled cells (blue arrow); (**F**) in VV2, a few immunolabeled RG fibers, single GS^+^ accumulations of neuroepithelial cells in PVZ (red arrow) and a high concentration of GS–cells in PVZ (in the white square) were revealed; (**G**) quantitative ratio of GS^+^ cells in intact animals (control group) and at 3 days after traumatic injury to the telencephalon (*n =* 5 in each group, * *p* ≤ 0.05; ** *p* ≤ 0.01; significant difference from control groups). One-way ANOVA followed by the Student–Newman–Keuls post hoc test. (**H**) Percentage proportion between GS^+^/GS–cells in all areas of the pallium and subpallium of juvenile chum salmon on day 3 posy-injury.

**Figure 5 ijms-22-01279-f005:**
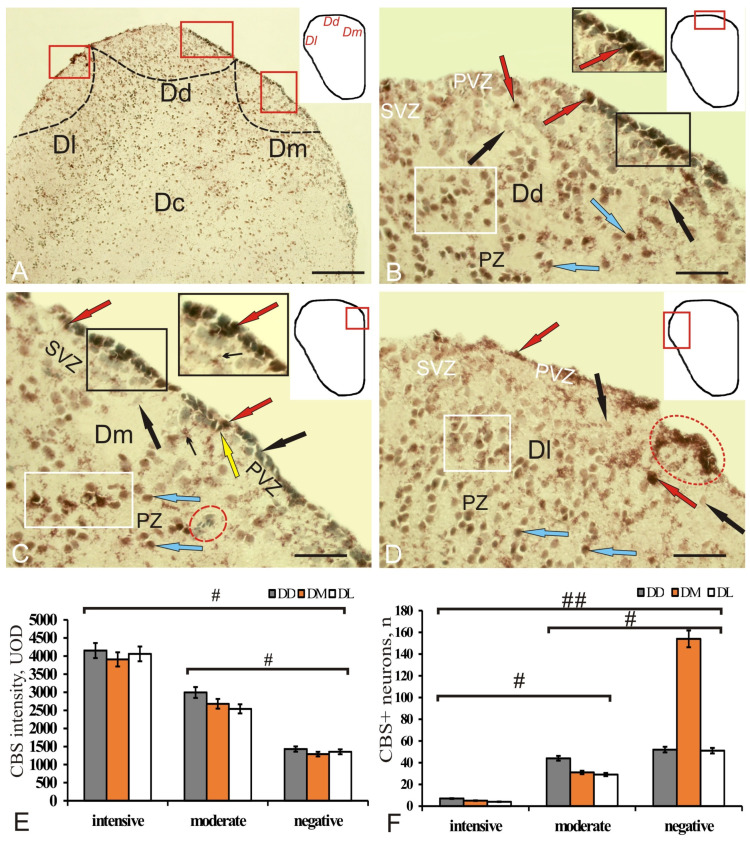
Cystathionine β-synthase (CBS) in the pallium of intact juvenile chum salmon, *Oncorhynchus keta*. (**A**) General view of the pallial region of the telencephalon, the pictogram shows the zones of the dorsal telencephalon (pallium), Dd—dorsal, Dm—medial, Dl—lateral, (outlined with red rectangles) black dotted lines indicate the boundaries of the pallial zones; (**B**) dorsal pallial zone (Dd) at higher magnification, inset (in black rectangle) shows a fragment including PVZ and SVZ, intensely labeled CBS^+^ neuroepithelial cells (red arrows), moderately labeled CBS cells (blue arrow), and negative cells (black arrow) in all areas; (**C**) medial pallial zone (Dm), some CBS^+^ cells showed proximal areas of radially directed processes (yellow arrow), clusters of small moderately CBS–labeled cells were detected in the SVZ (inset), and small clusters of microglia were revealed in PZ (in the red oval) and groups of intensely and moderately labeled cells (in a white rectangle); (**D**) lateral pallial zone (Dl) with a diffuse pattern of CBS-immunolocalization, aggregation of CBS^+^ cells in PVZ (in the red oval). Immunohistochemical labeling of cystathionine β-synthase. Scale bars: (**A**) 100 µm, (**B**–**D**) 50 µm. (**E**) Comparative densitometric analysis of CBS^+^ cells’ activity; #—significant intergroup differences (*p* < 0.05), in groups of intensely, moderately labeled cells and immunonegative cells (*n =* 5 in each group). One-way ANOVA. Intensity of CBS labeling, UOD. (**F**) Comparative distribution of CBS^+^ cells in DD, DM and DL; significant intergroup differences # (*p* < 0.05), ## (*p* < 0.01) between the groups of intensively, moderately labeled cells, and immunonegative cells (*n =* 5 in each group). One-way ANOVA.

**Figure 6 ijms-22-01279-f006:**
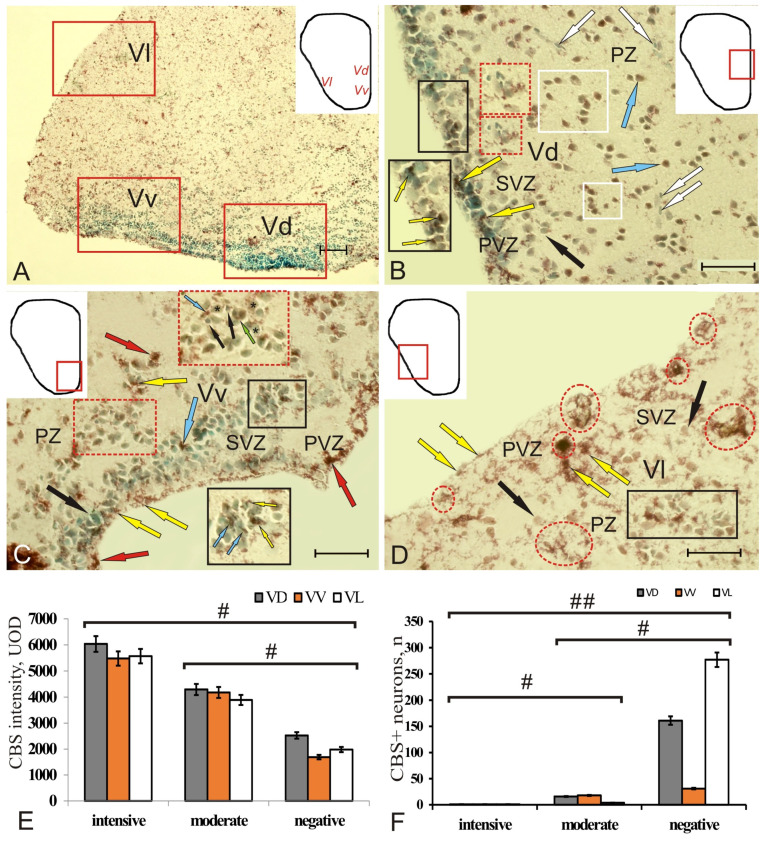
Cystathionine β-synthase (CBS) in the subpallium of intact juvenile chum salmon, *Oncorhynchus keta*. (**A**) General view of the subpallial region of the telencephalon, the pictogram shows the zones of the ventral telencephalon (subpallium), Vd—dorsal, Vv—ventral, Vl—lateral, (outlined with red rectangles) areas; (**B**) dorsal subpallial zone (Vd) at higher magnification, the inset (in the black rectangle) shows CBS^+^ granules in PVZ and SVZ (yellow arrows), a fragment (red inset) including CBS–cells surrounded by an aggregation of small elongated basophilic cells (white arrows), clusters of CBS^+^ cells in PZ (in a white rectangle); (**C**) ventral subpallial region (Vv), CBS^+^ granules (yellow arrows) on the apical surface of PVZ neuroepithelial cells and clusters in SVZ (black inset), elongated moderately labeled CBS cells and granules inside and on the surface of CBS cells (red inset), nuclei stained with methyl green are indicated by an asterisk, a green arrow indicates the figure of mitosis; (**D**) the lateral subpallial region (Vl) of the accumulation of small, intensely labeled cells (in the red oval); multiple CBS^+^ granules (in the black rectangle) were located in PZ on the surface of CBS cells. Immunohistochemical labeling of cystathionine β-synthase. Scale bars: (**A**) 100 µm, (**B**–**D**) 50 µm. (**E**) Comparative densitometric analysis of CBS^+^ cells’ activity; #—significant intergroup differences (*p* < 0.05) between groups of intensely, moderately labeled cells, and immunonegative cells (*n =* 5 in each group). One-way ANOVA. Intensity of CBS labeling, UOD. (**F**) Comparative distribution of CBS^+^ cells in VD, VV, and VL; significant intergroup differences # (*p* < 0.05), ## (*p* < 0.01) between the groups of intensively, moderately labeled cells, and immunonegative cells (*n =* 5 in each group). One-way ANOVA.

**Figure 7 ijms-22-01279-f007:**
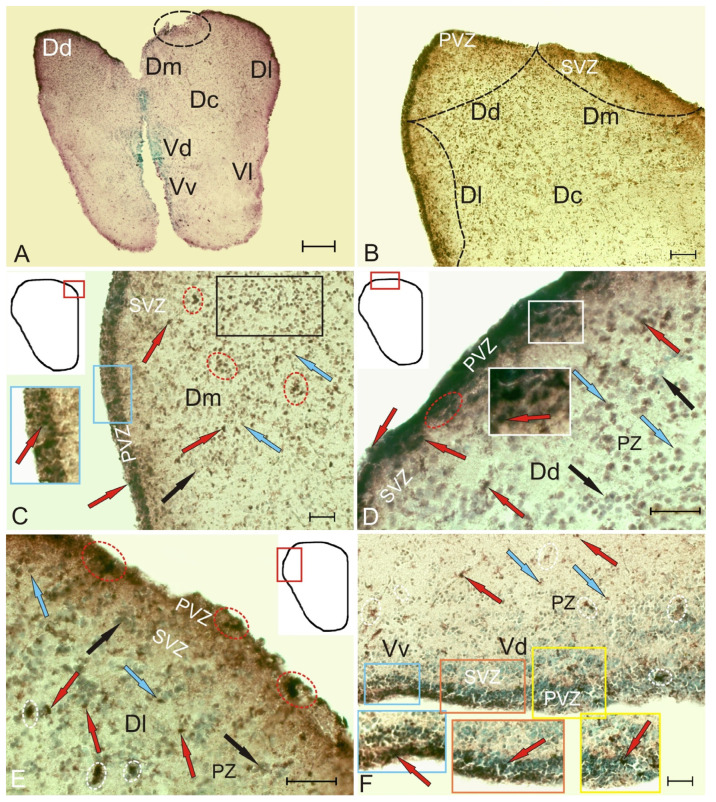
Cystathionine β-synthase in the pallium and subpallium of juvenile chum salmon, *Oncorhynchus keta*, at 3 days after the traumatic injury to the telencephalon. (**A**) General view of CBS immunolocalization in the telencephalon, where the black dotted line indicates the area of injury; (**B**) general view of the pallial zone, where black dotted lines indicate the boundaries of the pallial subzones; (**C**) small clusters of CBS^+^ cells in PVZ and SVZ (inset), CBS^+^ cells in PZ located discretely (red arrows) or forming small clusters (in red ovals), a large accumulation of CBS^+^ cells in PVZ (in the black rectangle); (**D**) the number of heterogeneous CBS^+^ cells and their aggregations (in the red oval) in SVZ increased in DD (inset); (**E**) clusters of CBS cells (black arrows), including moderately CBS^+^ cells (blue arrows), appeared in DL clusters containing intensely labeled cells in PVZ (in the red oval), in SVZ, clusters with intensely CBS-labeled cells appeared (in white ovals); (**F**) a layer of CBS^+^ cells occurred in the subpallium in VD in PVZ (yellow inset), SVZ (red inset), and VV (blue inset); single and paired CBS^+^ cells and their small clusters (in white ovals) in PZ. Immunohistochemical labeling of cystathionine β-synthase. Scale bars: (**A**) 500 µm; (**B**,**C**,**F**) 100 µm; (**D**,**E**) 50 µm.

**Figure 8 ijms-22-01279-f008:**
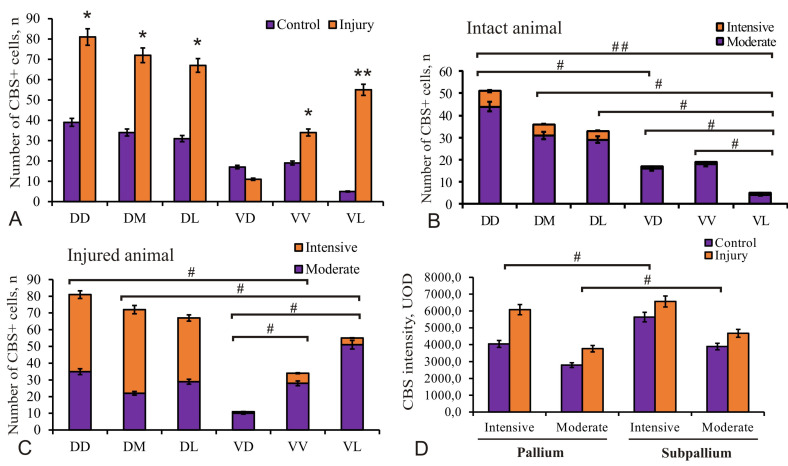
Quantitative ratio of CBS^+^/CBS–cells in the pallium and subpallium of intact juvenile chum salmon *Oncorhynchus keta* and at 3 days after the telencephalon injury. (**A**) Quantitative proportion of CBS^+^ cells in intact animals (control group) and in those at 3 days after the traumatic injury to the telencephalon (*n =* 5 in each group, * *p* ≤ 0.05; ** *p* ≤ 0.01; significant difference vs. control groups). One-way ANOVA followed by the Student–Newman–Keuls post hoc test. (**B**) Ratio of CBS^+^ intensely/moderately labeled cells in intact animals; (**C**) on day 3 post-injury, significant intergroup differences # (*p* < 0.05), ## (*p* < 0.01) between the groups of intensely, moderately labeled cells, and immunonegative cells (*n =* 5 in each group). One-way ANOVA. (**D**) Comparative densitometric analysis of the activity of CBS^+^ cells in the pallium and subpallium in intact animals and on day 3 days post-injury; significant intergroup differences # (*p* < 0.05), ## (*p* < 0.01) between the groups of intensely, moderately labeled cells, and immunonegative cells (*n =* 5 in each group). One-way ANOVA. Intensity of CBS labeling, UOD.

**Figure 9 ijms-22-01279-f009:**
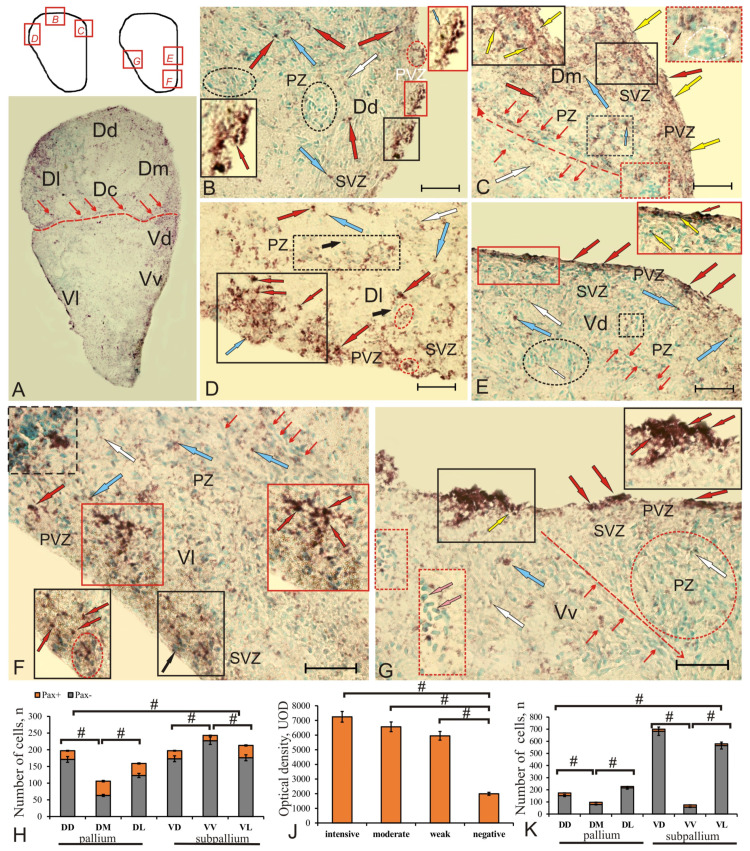
Pax2 expression in the pallium and subpallium of intact juvenile chum salmon, *Oncorhynchus keta*. (**A**) General view of the Pax2 expression pattern in the telencephalon of intact juvenile chum salmon where the boundaries between the pallial and subpallial regions are indicated by the red dashed line, Pax2^+^ cells along the border are indicated by thin red arrows; the pictograms indicate the areas of the brain shown in Figures (**B**–**D**) and (**E**–**G**). (**C**) Pax2 expression in DD, clusters of Pax2-expressing cells (red arrow) in PVZ (black inset) and nuclei (blue arrow, shown in red inset); the aggregation of nuclei is outlined by the red oval, and in black ovals there are aggregations of immunonegative cells in PZ, white arrows indicate weakly labeled cells; (**C**) patterns of Pax2 expression in DM; immunolabeling of Pax2 was detected in nuclei (shown by yellow arrows), which form extended regions in PVZ and SVZ (black inset). the granular pattern of Pax2 expression predominated in PZ, and the red dotted line indicates the radial direction of neuroblasts’ migration from constitutive neurogenic niches (CNN) (red inset) containing clusters of Pax2– (encircled by the white dotted line) and Pax2^+^ cells; (**D**) morphological heterogeneity of Pax2^+^ cells in DL, where the black rectangle outlines nuclei labeled with Pax2; Pax2^+^ cells were found in the basal part of PVZ and SVZ, both cellular (in the dashed rectangle) and nuclear expression of Pax2 (in the red ovals); (**E**) in PVZ of the dorsal subpallial region (VD), Pax2^+^ cells (red arrows) had a surface tangential orientation (red inset) and in SVZ and PZ there were moderately (in the dotted square) and weakly labeled Pax2 cells (in the oval), migrating cells (thin red arrows) and nuclei (yellow arrows) that form morphogenetic fields; (**F**) In VL, individual Pax2^+^ nuclei were encountered in PVZ (black inset); in SVZ, individual intensely labeled clusters of cells (red inset) were identified. In VL, local Pax2– cell clusters were identified in combination with dense clusters of Pax2^+^ cells (in dotted rectangle ); (**G**) intensely labeled clusters of Pax2^+^ cells (black inset) were detected in PVZ and the aggregation of immunonegative basophilic cells was found in PZ (indicated by pink arrows, red inset); the red dotted line indicates the direction of neuroblast migration and in the red oval there is a morphogenetic field with weak expression of Pax2 and neuroblasts. Immunohistochemical labeling of the protein product of the Pax2 transcription factor. Scale bars: (**A**) 500 µm; (**B**,**C**,**F**) 100 µm; (**D**,**E**) 50 µm. (**H**) Ratio of Pax2^+^/Pax2–cells in the pallium and subpallium of intact chum salmon (*n =* 5 in each group, #—significant intergroup differences (*p* < 0.05)). One-way ANOVA. (**J**) Comparative densitometric analysis of Pax2^+^ cells; significant intergroup differences # (*p* < 0.05) between groups of intensively, moderately, weakly labeled, and immunonegative cells (*n =* 5 in each group). One-way ANOVA. Intensity of Pax2 labeling, UOD. (**K**) Ratio of the number of Pax2^+^/Pax2—cells in the pallium and subpallium zones at 3 days after the injury to the telencephalon of juvenile chum salmon; #—significant intergroup differences (*n =* 5 in each group, #—significant intergroup differences (*p* < 0.05). One-way ANOVA.

**Figure 10 ijms-22-01279-f010:**
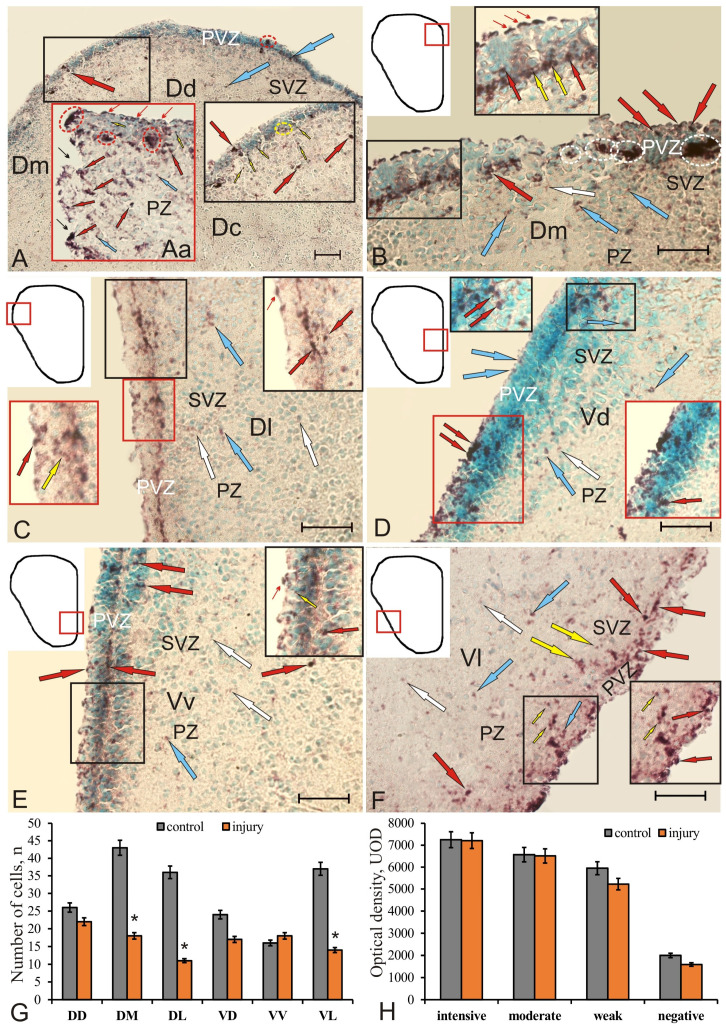
Pax2 expression in the pallium and subpallium of juvenile chum salmon, *Oncorhynchus keta*, at 3 days after the traumatic injury. (**A**) Pattern of Pax2 expression in DD, immunopositive cells (red arrows) and nuclei (yellow arrows) localized in PVZ and SVZ (in the black inset), where the aggregation of cells in PVZ is outlined by the red oval, nuclei (in the yellow oval); (**Aa**, in a red rectangle) in the area of injury, black narrow arrows indicate the traumatic lumen, RNN in red ovals; cells migrating to the area of injury are indicated by red narrow arrows, blue arrows indicate moderately labeled nuclei; (**B**) Pax2 expression in DM, migrating to the injury area, reactive cells in PVZ (red thin arrows, inset), local dense clusters of Pax2^+^ cells in the basal part of PVZ and SVZ (in white ovals), nuclei (yellow arrows); (**C**) Pax2 expression in DL, tangentially migrating cells in the apical and basal parts of PVZ (black inset), neuroepithelial cells (red arrow) and nuclei (yellow arrow in the red inset); (**D**) Pax2 expression in VD; Pax2^+^ nuclei and neuroepithelial cells were labeled with surface localization in PVZ (red inset) and SVZ (black inset), moderately labeled cells (blue arrows), weakly labeled cells (white arrows); (**E**) Pax2 expression in VV, numerous Pax2^+^ cells and nuclei were identified in the apical and basal parts of PVZ (inset); in SVZ and PZ, individual moderately (blue arrows) and weakly labeled Pax2 (white arrows) cells and nuclei prevailed; (**F**) in VL, an increased distribution of Pax2^+^ cells and their accumulations in the surface and basal parts of PVZ (inset), designations as in E. Immunohistochemical labeling of the protein product of the transcription factor Pax2. Scale bars: (**A**) 100 µm; (**B**–**F**) 50 µm. (**G**) Quantitative ratio of Pax2^+^ cells in intact animals (control group) and at 3 days after the traumatic injury to the telencephalon (*n =* 5 in each group, * *p* ≤ 0.05; significant difference vs. control groups). One-way ANOVA followed by the Student–Newman–Keuls post hoc test. (**H**) Comparative densitometric analysis of Pax2^+^ and Pax2– cells’ activity in intact juvenile chum salmon and at 3 days after the telencephalon injury (M ± SD). Intensity of Pax2 labeling, UOD.

## Data Availability

Data are available in the Appendix A.

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
