# Peer review of "Mechanical Brain Injury Increases Cells’ Production of Cystathionine β-Synthase and Glutamine Synthetase, but Reduces Pax2 Expression in the Telencephalon of Juvenile Chum Salmon, Oncorhynchus keta"

_ijms, 2021, doi:10.3390/ijms22031279_

Round 1
Reviewer 1 Report
Comments to the previous version:
Dear author, I have carefully evaluated the manuscript which I find to be extremely interesting. The topic treated is extremely fascinating, the evaluation of the alterations induced by traumatic events at the neuronal level, at the molecular level, is a fundamental prerequisite for being able to better understand the evolutionary phenomena that involve our organism. The work is drawn up in an exhaustive, extremely articulated way, but in some components this can make it more difficult to understand what is expressed. I consider it appropriate to describe in detail the background that led to the development of the study, the assumptions that led to its development. I would suggest making the study carried out easier to understand in order to make it accessible to anyone interested in this area. I would also add as the potential value of understanding the events observed in order to underline the scientific value of this type of study.
Author Response
Dear reviewer, we thank you for the time and effort that you have spent in reviewing our work. Thank you for your appreciation of our manuscript. Your earlier comments regarding the description of the prehistory that led to the development of the study were taken into account by us. Corresponding additions were made in page 2 line 85 ~ page 3 line 99:
Several key substances induce neurogenesis and reparative processes, with H2S being a candidate for the regulation of intercellular interactions performing the common cytoprotective and signal functions [13, 14]. It is a neuromodulator considered to be similar in functions to other gaseous compounds such as nitrogen oxide (NO) and carbon monoxide (CO). H2S is produced by several pathways, in particular by cystathionine β-synthase (CBS) which is a key enzyme in the formation of cysteine from methionine. This gasotransmitter activates intracellular signaling pathways through the sulfhydration of proteins and can react with protein hemes. There are multiple interplays between H2S and other gasotransmitters both on the level of their generation and as targets [15]. Previous studies on traumatic injury to the cerebellum in juvenils of masu salmon Oncorhynchus masou demonstrated sharply increasing the number of CBS+ cells after 3 days, which indicates the involvement of H2S in the post-traumatic response [16]. Similar results we observed after optic nerve injury in trout, showed a significant increase in the number of H2S-producing cells in the integrative centers of the brain: telencephalon, optic tectum, and cerebellum [17]. A noteworthy finding was the presence of CBS-producing radial glia in the optic tectum of trout after the optic nerve injury [18].
and in page 3 line 119 - 127:
Studies on trout have shown that a damage to the optic nerve leads to an increase in the number of Pax2+ reactive astrocytes in it, being involved in the initial stages of the optic nerve axon regeneration [26]. In the case of optic nerve injury, a significant increase in the number of Pax6+ cells has been revealed in the parts of the trout brain that have directed retinal inputs (the visual nuclei of the diencephalon and the optic tectum) [27]. It has found that some of Pax6+ cells have a neuroepithelial phenotype and are part of reactive neurogenic niches located in the periventricular zone (PVZ) and parenchymal regions of the brain. Another population of Pax6+ cells has a radial glia phenotype and arises as a result of activation of constitutive neurogenic domains, as well as within newly formed reactive neurogenic niches [26].
In accordance with your earlier comments, a Conclusions section was added to the manuscript, summarizing and summarizing the main content of the manuscript.

Reviewer 2 Report
In this article, the authors study a preclinical model of post-traumatic brain injury (TBI) recovery and reparative neurogenesis in the telencephalon of the juvenile chum salmon (Oncorhynchus keta). The authors mentioned Hydrogen sulfide (H2S), as one of the main neuromodulators produced by cystathionine ß-synthase (CBS) pathways The authors claimed that H2S has a key role to induce neurogenesis and reparative processes, increasing the number of CBS positive cells 3 days post-TBI. Specifically, the authors proposed that TBI increased the production of CBS and glutamine synthetase (GS), but reduces Pax2 expression in neurogenic areas and non-neurogenic parenchymal zones of the telencephalon (pallium and subpallium regions). The article brings to light an interesting topic with novel and relevant information indicating the potential neuroprotective effects of GS promoting an effective reparative neurogenesis. The studies described were conducted with the proper authorization of the institutional ethics committee. The article is well-written, and the study design and statistical analysis are straightforward. The material and methods are fairly well described and results clearly presented supporting rational conclusions. However, a moderate number of concerns need to be addressed before this article can move forward.
Moderate Concerns:
1. Although there is an overall idea of the multiple and complex molecular events and their roles, it is difficult to grasp which are the specific aims of this study. Maybe the authors can reorganize their ideas and provide a clearer picture of what they want to achieve. Short and specific highlighting points at the end of the introduction section could be helpful.
2. The interlink between many post-TBI molecular and structural elements described are only a small fraction of a far more complex mechanism of neurodegeneration (Gatto, RG 2020; Journal of Integrative Neuroscience). However, in what manner each of those biomarkers and mechanism described in this work can be linked is not completely clear, and sometimes confusing. Maybe the authors can illustrate such molecular pathways and their respective anatomical locations by presenting a diagram for illustrative purposes.
3. In the abstract section, the authors mentioned: “We believe that the decrease in the expression of Pax2 may be caused by the inhibitory effect of the Pax6 transcription factor, whose expression in the juvenile salmon brain increases upon injury”. However, the role of Pax6 is not supported by experimental data or presented in the results section. Are the authors referring to previous work on this biomarker (Pax6)? Why do the authors think this marker has such a different profile among other animal species (chicken, mammals)?
4. In the materials and methods section, how the authors calculated the minimal number of animals needed to reach statistically valid conclusions? Did the data showed any differences based on the (genetic) gender of the animals? (Schaeffer, L et al. 2018; Journal of Animal Breeding and Genetics).
5. The current study is only focused on the early post-TBI response (less than 3 days). As such, other molecular phenomena related to chronic responses are not considered in this work as the authors imply an absence of chronic TBI elements in this species. Could you provide any references to support this claim?
6. From the biological perspective, seems that the more likely natural TBI occurrence in this particular model is more related to mild & repetitive blunt TBI type scenario (rather than a unique perforating injury). As such, maybe the authors can briefly discuss the advantages and disadvantages of this model compared to other TBI models, such as controlled cortical impact (CCI)(Chauhan NB, et al. 2010; Journal of Neuroscience Methods & Gatto, RG et al. 2015; Restorative Neurology and Neuroscience), weight drop, fluid percussion injury, penetrating ballistic-like brain injury, blast injury, etc.
7. In terms of the injury’s severity, how the authors classify the degree of injury for quantitative purposes (mild, moderate, or severe)? A brief comment can suffice.
8. Did the authors observe any overlapping (double labeling) in these immunohistochemical (IHC) markers (as an example, Pax2 & GS or CBS & GS). As an example, (using DAB and Vector Red) that could suggest the presence of multiple & intersecting molecular mechanisms among the structures/cell investigated?
9. As a whole, the investigations presented only a histochemical evaluation of post-TBI potential biomarkers related to the GBS and GS pathways. Are the authors planning to use a more comprehensive biochemical analysis to validate their IHC results?
10. Please explain in more detail the criteria & calculations used to catalog the intense, moderate, weak (figure 9 and 10), and negative immunolabeling groups (discussed in lines 833). It is based on any specific Optical Density threshold number? Please note that such criteria seem not clarified in the material and methods section.
11. Moreover, why in the comparative densitometric analysis of CBS+ cells, as an example, showed in figure 5E the immunonegative cells seems to have a sizable (1000-1500 UOD) intensity after a negative background control subtraction? A brief explanation will suffice.
12. The choice of statistical analysis seems confusing. In the statistical section, the authors mentioned the use of the Student–Newman–Ceules t-test. What is the difference between this test and the Student–Newman–Keuls (SNK) method or the Student–Newman–Keuls t-test mentioned along with the legends of figure 4,8,10? Also, is it a standalone t-test or an ANOVA post-hoc test? Please revise and provide a proper & simple explanation for such disparities.
13. Is not clear what are the future directions of this study. Do the investigators aim to provide a better understanding of the mechanisms involved in post-TBI neurorepair processes for this fish only? Considering the previously mentioned differences in changes of these biomarkers in mammals (line 972-974), how such molecular targets can be potentially useful to test therapeutic targets in the TBI patient population?
Minor concerns:
14. In the abstract section, line 24: please include the acronym for Hydrogen Sulfite (H2S) when it is used for the first time.
15. For consistency and for readers not familiar with the topic, please clarify all the used acronyms when used for the first time (Shh, Sonic Hedgehog), (Pax2, Paired Box 2), etc. Conversely, the explanation of other acronyms has been repeated many times along the text (as an example, CBS in lines 89 and 143, etc.). Please be consistent.
16. In the intro section, line 121, please delete the extra space in the sentence: “...In the case of optic nerve injury...”.
17. In the materials and methods section, page 4, line 170-171, please rephrase the sentence: “To remove blood from the anesthetized animals, ...”
18 . Also, in line 171, please correct “parapharmaldehyde (PFA)…” to “paraformaldehyde”.
19. For reproducibility purposes, please add catalog and lot number for all chemicals used in the material and methods section (PFA, PBS, etc.).
20. In the statistical section, please revise:“… (ANOVA, Chicago, IL, USA) … “.
21. In the statistical section, the authors mentioned the use of ANOVA with Bonferroni correction nut in the legend of figure 2 the authors mentioned they used Student–Newman–Keuls post hoc test (see 12). Please clarify.
22. In the results section, is not clear the total number of cells analyzed (n=5)? or they refer to each cell count repeated on each animal (n=10)? As an example, the legend on figure 6E mentioned (n=5) cells per group but in the main text, the authors refer to the same figure (line 65) mentioned an (n=10). Please clarify and revise
22. In figure 8, please correct the label from 8C; “Ijured animal” to “Injured Animal”.
In sum, the study represents a valid attempt to continue the author's characterization of this model using immunohistochemical techniques. However, to move this important work forward, the careful address of each of the points listed above is highly encouraged.
Author Response
Dear reviewer, we thank you for a detailed analysis of our work and the expressed valuable comments that will significantly improve the content of our work. We deeply appreciate your helpful comments on our manuscript. We agree with the points that were raised, and have provided our responses to your comments below.
Moderate Concerns:
- Although there is an overall idea of the multiple and complex molecular events and their roles, it is difficult to grasp which are the specific aims of this study. Maybe the authors can reorganize their ideas and provide a clearer picture of what they want to achieve. Short and specific highlighting points at the end of the introduction section could be helpful.
The main purpose of the article was to study three regenerative-associated factors affecting constitutive and post-traumatic neurogenesis. An obvious connection between these factors has not yet been established, however, hydrogen sulfide and glutamine synthetase have a proneurogenic and neuroprotective effect. The study of the regulatory effect of Pax2 on constitutive and post-traumatic neurogenesis showed a more complex participation of this factor in the processes of post-traumatic neurogenesis, which has regional specificity.
- The interlink between many post-TBI molecular and structural elements described are only a small fraction of a far more complex mechanism of neurodegeneration (Gatto, RG 2020; Journal of Integrative Neuroscience). However, in what manner each of those biomarkers and mechanism described in this work can be linked is not completely clear, and sometimes confusing. Maybe the authors can illustrate such molecular pathways and their respective anatomical locations by presenting a diagram for illustrative purposes.
You are certainly right that the regenerative-associated factors investigated in our work are “a small part of a much more complex mechanism of neurodegeneration,” which is confirmed by the reviewer's reference in an authoritative publication. However, in our research, we were guided by a slightly different approach, taking into account the fact that the process of neuronal regeneration in fish and amphibians is generally successful in comparison with that in mammals. The factors that we studied in the brain of juvenile chum salmon using a simple TBI model have no connection at first glance. However, individually, these molecular markers, each in its own way, are involved in the process of neuronal regeneration. Reports on the involvement of GS in neuronal regeneration of the cerebellum of the Apteronotus (Zupanc, GK; Sîrbulescu, RF Teleost fish as a model system to study successful regeneration of the central nervous system. Curr. Top. Microbiol. Immunol. 2013, 367, 193–233) and Oncorhynchus masou (Pushchina, E.V.; Stukaneva, M.E.; Varaksin, A.A. Hydrogen sulfide modulates adult and reparative neurogenesis in the cerebellum of juvenile masu salmon, Oncorhynchus masou. Int. J. Mol. Sci. 2020, 21 (24), 9638) support this opinion. Our numerous, repeated observations on the presence of CBS in the matrix zones of the brain in juvenile salmonids were the main reason to study the participation of this factor in the TBI process. Currently, data on the effect of TF Pax2 are limited. Nevertheless, having some preliminary data from our own observations, we decided to test the dynamics of PAX2 participation in the TBI process. In this work, we presented the results obtained in the study of the dynamics of these molecular markers in the acute post-traumatic period, which had not been previously studied. We hope that further studies will provide an opportunity to clarify in more detail the relationship between the various signaling pathways in which these markers are involved, which will make it possible to present a clear diagram of the interaction between these factors. However, for now we are ready to present and discuss only the data already known from the literature and the results of our own observations about the participation of these molecular factors in the process of neuronal regeneration.
- In the abstract section, the authors mentioned: “We believe that the decrease in the expression of Pax2 may be caused by the inhibitory effect of the Pax6 transcription factor, whose expression in the juvenile salmon brain increases upon injury”. However, the role of Pax6 is not supported by experimental data or presented in the results section. Are the authors referring to previous work on this biomarker (Pax6)? Why do the authors think this marker has such a different profile among other animal species (chicken, mammals)?
Yes, of course, we are referring to previous work in which both factors were investigated in the brains of trout (26, 27). In the Discussion section, which suggests a possible mechanism for interaction between Pax2 and Pax6, it was said that “Our results are consistent with the data on immunofluorescent labeling that have shown the Pax2 distribution in the retina of adult guinea pigs, chickens, and mice [59]”.
- In the materials and methods section, how the authors calculated the minimal number of animals needed to reach statistically valid conclusions? Did the data showed any differences based on the (genetic) gender of the animals? (Schaeffer, L et al. 2018; Journal of Animal Breeding and Genetics).
According to ethical rules, the number of animals used for experimental manipulations should be minimal, but at the same time, their number should provide the required sampling rate to achieve a statistical power ≥ 0.95. According to the methods of statistical calculation carried out in works on Danio (Zambusi et al., 2020. Granulins Regulate Aging Kinetics in the Adult Zebrafish Telencephalon), the minimum number of animals to achieve a statistical power ≥0.95 is 4 animals. In our work, to form experimental groups, 5 animals were used for each marker, including intact and post-traumatic ones; in total, 30 animals were used (a typo in the Experimental animals section was corrected). The study was carried out on one-year-old juveniles of chum salmon, in which sex is not determined at this age, since the sex glands are not yet formed.
- The current study is only focused on the early post-TBI response (less than 3 days). As such, other molecular phenomena related to chronic responses are not considered in this work as the authors imply an absence of chronic TBI elements in this species. Could you provide any references to support this claim?
In this work, we focused on the study of IHC production of GS, CBS and Pax2 during the early response (3 days) after TBI. Previous studies of traumatic eye injury in trout after 1 week and IHC determination of CBS in the cells of the integrative centers of the brain: telencephalon, tectum and cerebellum showed an increase in the number of CBS + cells in all parts of the brain (Pushchina, EV; Varaksin, AA; Obukhov, DK; Prudnikov, IM GFAP expression in the optic nerve and increased Н2S generation in the integration centers of the rainbow trout (Oncorhynchus mykiss) brain after unilateral eye injury.Neural Regen Res. 2020, 15 (10), 1867-1886.doi: 10.4103 / 1673- 5374.280320; Pushchina, EV; Varaksin, AA; Obukhov, DK Cystathionine β-synthase in the brain of the trout Oncorhynchus mykiss after unilateral eye damage and in conditions of in vitro cultivation.Russian Journal of Developmental Biology. 2019, 50 (2), 39-58). These data were also confirmed by Western blotting and ELISA.
- From the biological perspective, seems that the more likely natural TBI occurrence in this particular model is more related to mild & repetitive blunt TBI type scenario (rather than a unique perforating injury). As such, maybe the authors can briefly discuss the advantages and disadvantages of this model compared to other TBI models, such as controlled cortical impact (CCI)(Chauhan NB, et al. 2010; Journal of Neuroscience Methods & Gatto, RG et al. 2015; Restorative Neurology and Neuroscience), weight drop, fluid percussion injury, penetrating ballistic-like brain injury, blast injury, etc.
Thank you for your valuable suggestion. Indeed, in our work, juveniles of chum salmon were inflicted with a common stab injury of the cerebral hemisphere according to the method of Kishimoto et al. 2012. In the literature on zebrafish there is evidence of chemical destruction of the telencephalon by acid injection, leading to a chemical burn of the telencephalon (Kyritsis, N .; Kizil, C .; Zocher, S .; Kroehne, V .; Kaslin, J .; Freudenreich , D .; Iltzsche, A .; Brand, M. Acute inflammation initiates the regenerative response in the adult zebrafish brain. Science. 2012, 2338, 1353-1356). In this work, the assessment of the proliferative post-traumatic response in the telencephalon after trauma was carried out. In our work, we aimed for a simple and controlled application of damage to the telencephalon, which can be easily reproduced by any researcher. In our previous studies on trout, we investigated unilateral traumatic eye injury with subsequent assessment of the dynamics of regenerative-associated factors (Pushchina, EV; Varaksin, AA; Obukhov, DK; Prudnikov, IM GFAP expression in the optic nerve and increased Н2S generation in the integration centers of the rainbow trout (Oncorhynchus mykiss) brain after unilateral eye injury.Neural Regen Res. 2020, 15 (10), 1867-1886; Pushchina, EV; Varaksin, AA Neurolin expression in the optic nerve and immunoreactivity of Pax6-positive niches in the brain of rainbow trout (Oncorhynchus mykiss) after unilateral eye injury.Neural Regen Res. 2019, 14 (1), 156-171.). In this case, the trauma was not direct, and the assessment of the cellular response was performed in various parts of the trout brain. Perhaps, in our future studies, we will try to modify direct traumatic injury to various parts of the brain and compare data from stab wound trauma, which is the "gold standard" of TBI in animal models.
- In terms of the injury’s severity, how the authors classify the degree of injury for quantitative purposes (mild, moderate, or severe)? A brief comment can suffice.
For the telencephalon of juvenile chum salmon, this type of injury can be considered a mild injury. The conducted behavioral observations show an insignificant change in behavioral activity and 100% survival of animals, compared with a similar injury to other parts of the brain - the cerebellum or the mesencephalic tegmentum. When these parts of the brain are injured in animals, behavioral activity changes significantly, for example, rotational movements appear after injury of the cerebellum.
- Did the authors observe any overlapping (double labeling) in these immunohistochemical (IHC) markers (as an example, Pax2 & GS or CBS & GS). As an example, (using DAB and Vector Red) that could suggest the presence of multiple & intersecting molecular mechanisms among the structures/cell investigated?
We plan to experiment with double labeling in the near future.
- As a whole, the investigations presented only a histochemical evaluation of post-TBI potential biomarkers related to the GBS and GS pathways. Are the authors planning to use a more comprehensive biochemical analysis to validate their IHC results?
Yes, of course, IHC labeling is planned to be combined with Western immunoblotting, enzyme-linked immunosorbent assay and double immunofluorescence labeling.
- Please explain in more detail the criteria & calculations used to catalog the intense, moderate, weak (figure 9 and 10), and negative immunolabeling groups (discussed in lines 833). It is based on any specific Optical Density threshold number? Please note that such criteria seem not clarified in the material and methods section.
In this work, a densitometric assessment of the optical density of immunolabeled cells was carried out using the Wizard software. The primary densitometric data were grouped according to the qualitative definition of immunolabeling. As a result, several groups (3 or 4, respectively) were obtained with a ranged OD value in each group. Background optical density was measured in 5 independent sites of each brain preparation and averaged. Further, from the averaged background OD value, the average OD value for each group (intense, moderate, and weak (for Pax2) was subtracted. Cells without IHC signal stained with methyl green were considered immunonegative. Since the intensity of immunolabeling for different markers GS, CBS Pax2, respectively, the background differed, for each marker, quantitative values for intensely, moderately and weakly labeled cells were calculated individually in accordance with specific quantitative indicators of the IHC signal and background.
- Moreover, why in the comparative densitometric analysis of CBS+ cells, as an example, showed in figure 5E the immunonegative cells seems to have a sizable (1000-1500 UOD) intensity after a negative background control subtraction? A brief explanation will suffice.
In a comparative densitometric analysis of the OD of the pallial zones, the immunonegative cells did not have an IP signal, but were stained with MG. The OD of cells when stained with MG was 1000-1500 UOD.
- The choice of statistical analysis seems confusing. In the statistical section, the authors mentioned the use of the Student–Newman–Ceules t-test. What is the difference between this test and the Student–Newman–Keuls (SNK) method or the Student–Newman–Keuls t-test mentioned along with the legends of figure 4,8,10? Also, is it a standalone t-test or an ANOVA post-hoc test? Please revise and provide a proper & simple explanation for such disparities.
One-way analysis of variance (ANOVA), followed by the Student-Newman-Keuls post hoc test, was used to determine significant differences in control animals and those subjected to TBI
- Is not clear what are the future directions of this study. Do the investigators aim to provide a better understanding of the mechanisms involved in post-TBI neurorepair processes for this fish only? Considering the previously mentioned differences in changes of these biomarkers in mammals (line 972-974), how such molecular targets can be potentially useful to test therapeutic targets in the TBI patient population?
We believe that the increased production of hydrogen sulfide and glutamine synthetase on the one hand in the brain of juvenile chum salmon provides the necessary background for a favorable course of repair processes. In the mammalian brain, as a result of TBI, both substances are synthesized in insufficient quantities, in particular, the CBS level after a short-term increase decreases rapidly (Zhang, M .; Shan, H., Wang, T., Liu, W., Wang, Y., Wang, L., Zhang, L., Chang, P., Dong, W., Chen, X., Tao, L. Dynamic change of hydrogen sulfide after traumatic brain injury and its effect in mice.Neurochem.Res. 2013, 38, 714-725). We believe that the therapeutic link in this case could be the injection of slowly released H2S donors, which has antioxidant, neuroprotective, and antiapoptotic effects. Regarding GS, which is an enzyme that converts toxic glutamate into neutral glutamine and, at the same time, is a molecular marker of NSC in the vertebrate brain, the situation is somewhat more complicated. The data obtained by us make it possible to clarify that in intact juvenile chum salmon in the telencephalon, NSCs of the embryonic type with a neuroepihelial phenotype prevail. After TBI, the brain of juvenile chum salmon activates the aNSC population with the radial glia phenotype. The data obtained allow us to conclude that in juvenile chum salmon there is an alternative / additional cellular mechanism, which consists in the activation of aNSC, along with eNSC, which enhances the reparative potential of the brain during trauma. Further elucidation of the neurogenic and neuroprotective role of GS and H2S is required to decipher the mechanisms of successful brain regeneration in juvenile salmon.
Minor concerns:
- In the abstract section, line 24: please include the acronym for Hydrogen Sulfite (H2S) when it is used for the first time.
included
- For consistency and for readers not familiar with the topic, please clarify all the used acronyms when used for the first time (Shh, Sonic Hedgehog), (Pax2, Paired Box 2), etc. Conversely, the explanation of other acronyms has been repeated many times along the text (as an example, CBS in lines 89 and 143, etc.). Please be consistent.
clarifed
- In the intro section, line 121, please delete the extra space in the sentence: “...In the case of optic nerve injury...”.
deleted
- In the materials and methods section, page 4, line 170-171, please rephrase the sentence: “To remove blood from the anesthetized animals, ...”
rephrased
18 . Also, in line 171, please correct “parapharmaldehyde (PFA)…” to “paraformaldehyde”.
corrected
- For reproducibility purposes, please add catalog and lot number for all chemicals used in the material and methods section (PFA, PBS, etc.).
added
- In the statistical section, please revise:“… (ANOVA, Chicago, IL, USA) … “.
revised
- In the statistical section, the authors mentioned the use of ANOVA with Bonferroni correction nut in the legend of figure 2 the authors mentioned they used Student–Newman–Keuls post hoc test (see 12). Please clarify.
clarifed
- In the results section, is not clear the total number of cells analyzed (n=5)? or they refer to each cell count repeated on each animal (n=10)? As an example, the legend on figure 6E mentioned (n=5) cells per group but in the main text, the authors refer to the same figure (line 65) mentioned an (n=10). Please clarify and revise
n=5 refer to each cell count repeated on each animal
- In figure 8, please correct the label from 8C; “Ijured animal” to “Injured Animal”. In sum, the study represents a valid attempt to continue the author's characterization of this model using imunohistochemical techniques. However, to move this important work forward, the careful address of each of the points listed above is highly encouraged.
corrected

This manuscript is a resubmission of an earlier submission. The following is a list of the peer review reports and author responses from that submission.
Round 1
Reviewer 1 Report
The authors insist when the brain was injured such as TBI, H2S related molecules including CBS and GS were activated but Pax2 was decreased. However, there was no direct evidence that H2S molecule was up-regulated by TBI. The title of this manuscript was H2S-dependent modulation involved in decreasing of post-traumatic glutamine excitotoxicity, however, their results could not convince their conclusion sufficiently since there is no direct evidence of up-regulation of H2S.
Immuno-hisochemical sections showed that CBS and GS positive cells were increased. Unfortunately, they did not show low magnified image of telencephalon which show the distribution of positive cell throughout the whole telencephalon as Figure 9 (A) which show the disribution of PAX2 positive cells throughout the whole telencephalon.
The authos insisted that intensive positive cells were present in the outermost area of sections, however, these signals would be artifacts.
In addition, the abstract section is like introduction section. It should be more comprehensive which include a clear statement of the problem addressed, the aims and objectives, pertinent results, conclusions from their study.
Reviewer 2 Report
The manuscript has to be highly improved before considering publication. The figures, tables, and comments should be highly reduced so that the text is clearer and the final conclusions more understandable.
Although there is a significant load of work in this study, the conclusions remain highly speculative, and should be expressed as such.
Reviewer 3 Report
Dear author, I have carefully evaluated the manuscript which I find to be extremely interesting. The topic treated is extremely fascinating, the evaluation of the alterations induced by traumatic events at the neuronal level, at the molecular level, is a fundamental prerequisite for being able to better understand the evolutionary phenomena that involve our organism. The work is drawn up in an exhaustive, extremely articulated way, but in some components this can make it more difficult to understand what is expressed. I consider it appropriate to describe in detail the background that led to the development of the study, the assumptions that led to its development. I would suggest making the study carried out easier to understand in order to make it accessible to anyone interested in this area. I would also add as the potential value of understanding the events observed in order to underline the scientific value of this type of study.